# *D2E*: Scaling Vision-Action Pretraining on Desktop Data for Transfer to Embodied AI

**Suhwan Choi**[*]
MAUM.AI

**Jaeyoon Jung**[*]
MAUM.AI

**Haebin Seong**[*]
MAUM.AI

**Minchan Kim**
MAUM.AI

**Minyeong Kim**
Stanford University

**Yongjun Cho**
MAUM.AI

**Yoonshik Kim**
MAUM.AI

**Yubeen Park**
MAUM.AI

**Youngjae Yu**[†]
Seoul National University

**Yunsung Lee**[†]
MAUM.AI

## ABSTRACT

Large language models leverage internet-scale text data, yet embodied AI remains constrained by the prohibitive costs of physical trajectory collection. Desktop environments—particularly gaming—offer a compelling alternative: they provide rich sensorimotor interactions at scale while maintaining the structured observation-action coupling essential for embodied learning. We present **D2E** (Desktop to Embodied AI), a framework that demonstrates desktop interactions can serve as an effective pretraining substrate for robotics embodied AI tasks. Unlike prior work that remained domain-specific (e.g., VPT for Minecraft) or kept data proprietary (e.g., SIMA), D2E establishes a complete pipeline from scalable desktop data collection to verified transfer in embodied domains. Our framework comprises three components: (1) the OWA Toolkit that unifies diverse desktop interactions into a standardized format with 152× compression, (2) the Generalist-IDM that achieves strong zero-shot generalization across unseen games through timestamp-based event prediction, enabling internet-scale pseudo-labeling, and (3) VAPT that transfers desktop-pretrained representations to physical manipulation and navigation. Using 1.3K+ hours of data (259 hours of human demonstrations and 1K+ hours of pseudo-labeled gameplay), our 1B-parameter model achieves 96.6% success on LIBERO manipulation and 83.3% on CANVAS navigation, matching or surpassing models up to 7× larger, such as $\pi_0$ (3.3B) and OpenVLA (7B). These results demonstrate that sensorimotor primitives learned from digital interactions transfer effectively to real-world physical tasks, establishing desktop pretraining as a practical paradigm for embodied AI. All resources are publicly available at `https://worv-ai.github.io/d2e/`.

## 1 INTRODUCTION

Large-scale datasets have driven recent progress in large language models (LLMs) (Kaplan et al., 2020; Hoffmann et al., 2022), where pretraining on internet-scale resources enables strong generalization across diverse downstream tasks. In contrast, embodied AI has yet to experience such a scaling breakthrough. Unlike text, which can be collected from the web with minimum effort, embodied trajectories demand specialized hardware, costly human operation, and complex pipelines for annotation (Mandlekar et al., 2019; Qin et al., 2023; Fu et al., 2024; Cheng et al., 2024; Park et al., 2024). As a result, most existing datasets remain relatively small, domain-specific, and fragmented across incompatible formats (Geng et al., 2025), preventing the emergence of a true "data flywheel" for embodied AI.

Desktop interactions—screen, keyboard, and mouse—offer a compelling alternative to scale vision-action learning (Baker et al., 2022; Raad et al., 2024). These interactions are abundant, as desktop interfaces are well standardized and human-centric. Millions of users generate rich interaction trajectories through everyday digital activities. Crucially, desktop environments preserve the tight ob-

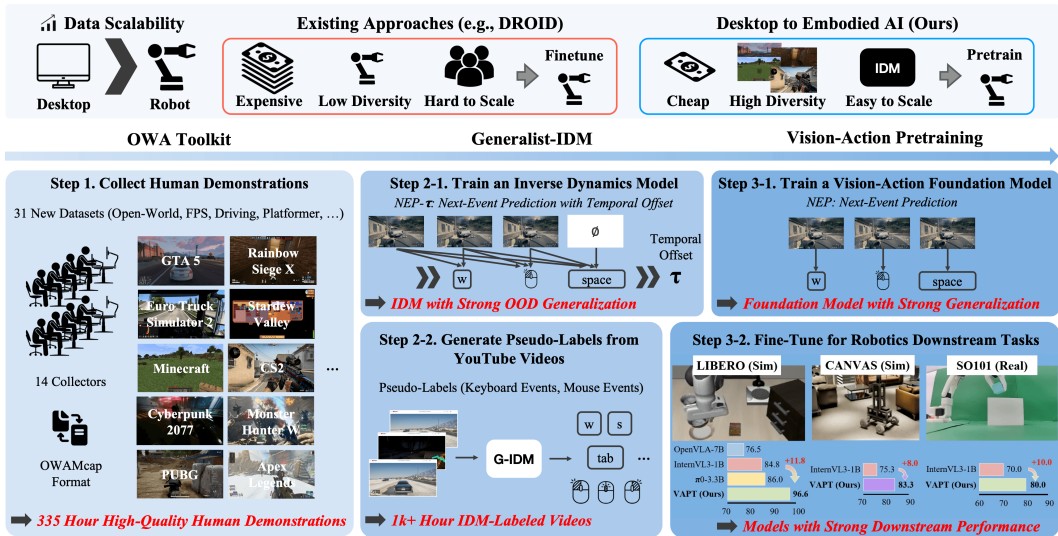

Figure 1: **Overview of D2E framework.** (1) The OWA Toolkit captures 335.6 hours of rich desktop demonstrations across 31 games, while the OWAMcap format achieves 152× compression compared to existing formats. (2) The Generalist-IDM uses next-event prediction with temporal offset (NEP-$\tau$) to achieve OOD generalization, enabling pseudo-labeling of 1K+ hours of YouTube gameplay. (3) Vision-Action Pretraining transfers desktop-pretrained representations to embodied AI, achieving 96.6% success on LIBERO manipulation and 83.3% on CANVAS navigation benchmarks which demonstrates desktop-to-robotics transfer.

servation–action coupling essential for embodied learning, while abstracting away hardware-specific constraints (Tang et al., 2025; Shridhar et al., 2020; Raad et al., 2024). Gaming interactions, in particular, exhibit complex sensorimotor patterns—navigation, object manipulation, and strategic planning—that mirror many embodied AI challenges, and are widely available at internet scale through shared gameplay videos.

We introduce **D2E (Desktop to Embodied AI)**, a framework that systematically transforms desktop interactions into a scalable pretraining substrate for embodied AI (Figure 1). D2E addresses two fundamental challenges: establishing a unified pipeline for high-quality desktop data collection and leveraging vast amounts of unlabeled internet videos beyond manual annotation.

Our first contribution, the **Open-World Agents (OWA) Toolkit**, provides the infrastructure for scalable desktop data capture. Built on Windows APIs and GStreamer (Microsoft Corporation; GStreamer Team), OWA's `ocap` recorder synchronizes multimodal streams—screen, keyboard, and mouse—into time-aligned events, while our OWAMcap format achieves order-of-magnitude compression improvements over existing formats. Through OWA, we collected 335 hours of human demonstrations across 31 diverse games and applications, establishing a foundation for desktop-based pretraining. Desktop data collection is substantially more cost-efficient than robotics data acquisition. 14 annotators collected our 335-hour corpus in one month, whereas DROID (Khazatsky et al., 2024) required 50 collectors across 13 institutions over 12 months to assemble a dataset of comparable scale.

Beyond human demonstrations, we introduce the **Generalist Inverse Dynamics Model (Generalist-IDM)** to demonstrate a pathway toward internet-scale data collection. By reformulating action prediction as timestamp-aware next-event prediction (NEP-$\tau$, introduced in Section 4.2), our model achieves strong zero-shot generalization—substantially outperforming specialist baselines on unseen games with minimal compute requirements. This generalization capability enables automatic pseudo-labeling of YouTube gameplay videos, expanding our dataset by over $1,000$ hours.

We demonstrate that desktop-pretrained representations transfer meaningfully to physical robotics through **Vision-Action PreTraining (VAPT)**. Models pretrained on our combined desktop corpus show consistent improvements on standardized benchmarks: It achieves a total success rate of 96.6% on *LIBERO* manipulation (Liu et al., 2023) and 83.3% on *CANVAS* navigation (Choi et al., 2024). These results establish, for the first time, that the sensorimotor patterns learned from desktop interactions can directly enhance performance in embodied AI domains, validating desktop data as a practical alternative to costly physical data collection.

Our contributions are threefold:

1. **OWA Toolkit**: A framework that contains `ocap` for synchronized event recording with FHD/QHD 60 Hz support, OWAMcap format for compact storage, and an optimized data pipeline for ML training—achieving up to $152\times$ compression and $41\times$ lower average disk read per image compared to TorchCodec; used to collect 335 hours of human demonstrations.

2. **Generalist-IDM**: An inverse dynamics model that outperforms game-specific Specialist IDMs, exhibiting out-of-domain generalization and in-context adaptation (e.g., calibrating mouse scale). Trained on OWA-collected data with around 192 H100-hours ($\sim$ \$800), the strong generalization of Generalist-IDM allows us to pseudo-label over 1K+ hours of YouTube gameplay.

3. **VAPT foundation model**: A vision-action pretrained model trained on 1.3K hours of desktop data from OWA and Generalist-IDM pseudo-labeling, transferring desktop knowledge to robotics. VAPT achieves 96.6% success on manipulation (*LIBERO*) and 83.3% on navigation (*CANVAS*).

## 2 RELATED WORK

**Collecting Data for Vision-Action Pretraining.** Large-scale vision-action (or vision-language-action) pretraining depends on multimodal corpora that pair perception with grounded actions across diverse tasks (Kaplan et al., 2020; Hoffmann et al., 2022). Recent embodied agents unify perception and control in a single model across heterogeneous domains (Reed et al., 2022; Firoozi et al., 2024; Wen et al.). In robotics, resources are emerging: RT-1 (Brohan et al., 2022) and RT-2 (Zitkovich et al., 2023) scale vision–language–action to real robots; Open X-Embodiment aggregates heterogeneous datasets to train RT-X models (O'Neill et al., 2024); and LeRobot (Cadene et al., 2024) lowers the barrier to collecting and reusing real-world datasets. Despite this progress, assembling real-robot interaction at meaningful scale remains challenging because of fragmented tooling, hardware overhead, and safety constraints (Xing et al., 2025; Park et al., 2024; Geng et al., 2025). Similarly, desktop interfaces lack open, standardized corpora and toolkits, bottlenecking vision-action pretraining (Tang et al., 2025; Chen et al., 2025). VPT (Baker et al., 2022) offers human-annotated and pseudo-labeled Minecraft trajectories but remains single-domain, while SIMA (Raad et al., 2024) demonstrates cross-game generalization through a unified interface yet keeps data proprietary. PLAICraft (He et al., 2025) advances multimodal Minecraft logging, but these efforts are environment-specific; broad cross-application generalization requires unified schemas that cover diverse desktop applications (McCarthy et al., 2025). Unlike prior single-domain or proprietary efforts, we contribute an open, unified, multi-game desktop-action dataset (31 games; 335ĥ) and an open-source toolkit, explicitly validated for transfer to embodied tasks.

**Inverse Dynamics Models.** Agents observe the states up to time $t-1$ and predict the action at time $t$. In contrast, Inverse Dynamics Models (IDMs) condition on surrounding states—past and future—to infer the action taken at time $t$. IDMs have been pivotal for scaling imitation learning to Internet-scale datasets, serving as pseudo-labelers for otherwise unlabeled action data (Ye et al., 2024; Bjorck et al., 2025). In robot manipulation, UniPi (Du et al., 2023) explores text-guided video generation to couple language grounding with policy learning, and LAPA (Ye et al., 2024) shows that latent action pretraining from videos can improve scalability and robustness. On the desktop side, VPT (Baker et al., 2022) trained a Specialist IDM on human-annotated Minecraft trajectories and used it to pseudo-label thousands of hours of Minecraft gameplay on YouTube. We demonstrate the potential of a Generalist-IDM, spanning multi-game, desktop-wide settings (McCarthy et al., 2025). Our design also differs from common tick-based IDMs (Baker et al., 2022; Ye et al., 2024), which fix a prediction window (e.g., 50 ms) and thus must emit a prediction each tick—inefficient in sparse-event regimes and coarse in temporal resolution. Instead, our IDM predicts the event *and* its timestamp, enabling event-driven modeling that avoids "no-op" ticks and makes more efficient use of inference context.

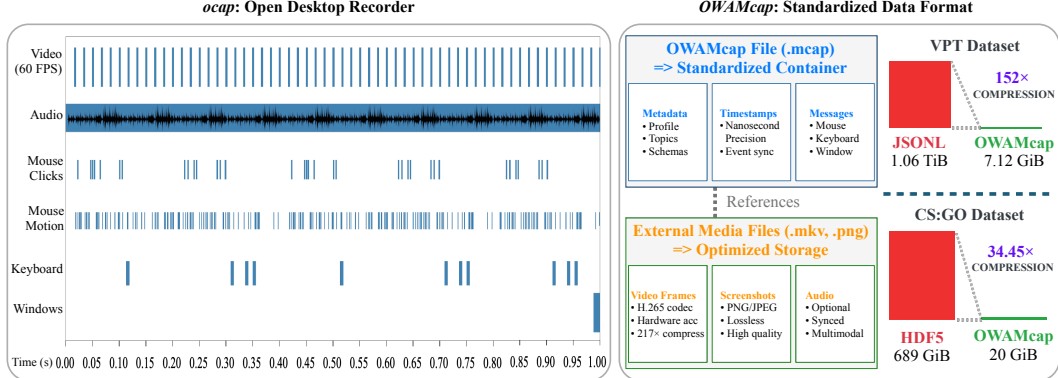

Figure 2: **OWA Toolkit's recording and storage architecture.** (Left) ocap recorder captures perfectly synchronized multimodal streams—video (60 FPS), audio, mouse events, keyboard inputs, and window states—with precise time alignment, enabling accurate reconstruction of desktop interactions. (Right) OWAM-cap format revolutionizes desktop data storage through its dual-layer architecture: standardized MCAP container for crash-safe metadata and event logging, paired with external media referencing for optimized video storage using H.265 codec (217× compression). This design achieves dramatic storage reduction—152× for VPT dataset (1.06 TiB → 7.12 GiB) and 34.45× for CS:GO dataset (689 GiB → 20 GiB)—while maintaining event fidelity and enabling efficient random access for training.

# 3 OPEN-WORLD AGENTS TOOLKIT

We introduce the **Open-World Agents (OWA) Toolkit** alongside large-scale desktop data, establishing both the infrastructure and data foundation for embodied AI research. The toolkit provides a unified interface (Zhang et al., 2024; 2025) for capturing interaction patterns across diverse applications without domain-specific action space definitions, while our data release demonstrates the practical scalability and diversity achievable through this standardized approach.

## 3.1 OCAP: SYNCHRONIZED DESKTOP RECORDER

Existing desktop recording tools lack critical features for desktop data collection. Content creation tools like OBS Studio (OBS Project) focus on streaming quality, while action modeling requires synchronized input event logging to capture the precise keyboard and mouse actions that caused visual changes. The ocap (Omnimodal CAPture) tool addresses this gap by capturing desktop signals in a synchronized manner, recording video, audio, keyboard, and mouse interactions with high temporal precision. Figure 2 (Left) illustrates an event timeline where these multimodal streams are well synchronized. By leveraging hardware acceleration using Windows APIs, we achieve real-time FHD/QHD recording at 60 Hz on consumer-grade GPUs with low overhead, ensuring that normal user activities remain unaffected and effectively lowering the hardware barrier for large-scale data collection. Implementation details are in Appendix A.

## 3.2 OWAMCAP: STANDARDIZED DATA FORMAT

Prior desktop datasets suffer from storage inefficiency and poor random access capabilities. Existing approaches (Baker et al., 2022; Pearce & Zhu, 2022) either store image-encoded frames in monolithic tables unsuitable for real-time recording, or use formats like JSONL that lack proper indexing and crash-safety. To address these limitations, we introduce OWAMcap (Figure 2, Right), which extends the industry-standard MCAP format (Foxglove, 2022)—widely adopted in robotics for multimodal sensor logging and providing efficient indexing, crash-safe writes, and broad ecosystem support—with two key desktop-specific additions.

First, we define standardized message schemas for desktop events (screen, keyboard, mouse) based on Windows APIs, enabling unified processing across different datasets without complex post-processing logic. Unlike other formats (e.g., RLDS (Ramos et al., 2021)) that lack solid message

definitions, our standardized schemas allow users to process identical message sets through a single pipeline for foundation model training.

Second, MediaRef enables efficient video storage while maintaining MCAP compatibility. Raw video captures and image encoding approaches like PNG are prohibitively large for foundation model training, making efficient compression essential. MediaRef addresses this by enabling modern video codecs (H.265), achieving $217\times$ compression over raw captures and $68\times$ over PNG while maintaining sufficient visual quality for agent training (Table 10).

## 3.3 OPTIMIZED DATA PIPELINE

Training foundation models on OWAMcap data requires specialized data loading strategies to maximize throughput, as I/O and data pipeline bottlenecks have been identified as critical limitations in large-scale video model training (Zhao & Krähenbühl, 2023; Leclerc et al., 2023). We present a four-stage optimized pipeline: (1) Media transcoding with optimized x264 parameters for consistent random access; (2) Event dataset conversion to HuggingFace datasets (Lhoest et al., 2021) format for efficient sequential and random access; (3) Fixed Sequence Length Dataset (FSLDataset) generation through tokenization and packing to maximize training throughput; (4) On-the-fly media loading with adaptive batch decoding that defers expensive media operations until training time. Our complete data pipeline optimizations are detailed in Appendix A.7, with comprehensive benchmark configurations provided in Appendix A.8.

**Fixed Sequence Length Dataset (FSLDataset)**  To optimize training throughput, we introduce FSLDataset that packs sequences to uniform lengths while preserving episode structure. Unlike conventional random concatenation, FSLDataset sequentially lists events within each episode up to the maximum sequence length, terminating at episode completion. This design enables consistent batch processing and converts fine-grained random access into coarse, coalesced patterns for improved I/O efficiency.

**Adaptive Batch Decoding Strategy**  Video decoding requires seeking to a keyframe and then sequentially decoding subsequent frames, since compressed video formats do not support independent decoding of arbitrary frames. To address this constraint, our adaptive batch decoding algorithm operates as follows: (1) seek to the target frame; (2) demux and decode until a keyframe is encountered; (3) upon hitting a keyframe, resume seeking to the target frame. This strategy ensures stable performance across fine-grained, coarse-grained, and mixed access patterns.

**Benchmarking Media Decoding on FSLDataset**  We evaluate our optimized pipeline on a representative FSLDataset containing 64 episodes of 5-minute Minecraft gameplay at 640×360 resolution and 20 Hz. The baseline uses single-frame decoding per frame, while TorchCodec and our approach use batch decoding for all frames within each FSLDataset sample. Throughput is measured as images processed per second, while I/O efficiency is measured as average disk read per image using isolated filesystem monitoring. Combining these optimizations—optimized x264 parameters and adaptive batch decoding—our complete pipeline achieves 119.16 img/s (10.2× over baseline) while reducing average disk read per image to 18.73 KB (3.4× less than baseline and 41× less than Torch-Codec (PyTorch Team, 2024)). Table 1 summarizes results across different configurations.

**InternVL3-1B Training Throughput**  Using the FSLDataset from the media decoding benchmark, we benchmarked InternVL3-1B training throughput on single H100 GPU. Our optimized pipeline achieves 4.77 it/s with 1 dataloading worker, while the baseline requires 16 workers to reach comparable throughput (4.55 it/s), demonstrating 16× efficiency gains. Moreover, the baseline performance saturates beyond 8 workers, indicating fundamental I/O bottlenecks that our optimizations successfully address (Table 2).

## 3.4 COLLECTING HUMAN DEMONSTRATIONS AT SCALE

We collect a desktop dataset that provides high-quality, synchronized multimodal signals for vision-action pretraining. While the OWA Toolkit can capture arbitrary desktop tasks (e.g., web surfing, productivity applications) with multimodal events—including the screen, mouse, and keyboard—we

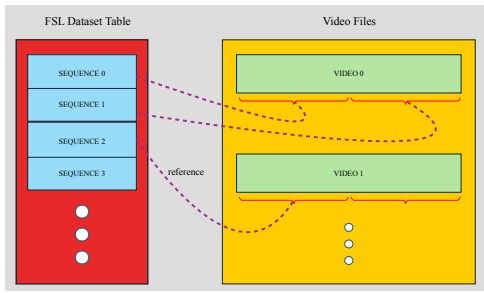

Figure 3: Our FSLDataset design, coupled with a batched decoding API, converts fine-grained random I/O into coarse, coalesced random access, thereby avoiding the limitations of large-scale filesystems that are inefficient for small random reads.

Table 1: Media decoding benchmark on FSL-Dataset (Minecraft, 64×5 min, 640×360 @ 20Hz).

| Configuration | Throughput (img/s) | Avg. Read (KB/img) |
|---|---|---|
| Baseline | 11.68 | 63.46 |
| + optimized x264 | 24.25 | 41.69 |
| + TorchCodec | 79.73 | 770.39 |
| **+ Ours** | **119.16** | **18.73** |

Table 2: InternVL3-1B training throughput on FSLDataset.

| Configuration | Throughput (it/s) |
|---|---|
| Ours (1 worker) | **4.77** |
| Ours (4 workers) | 4.73 |
| Baseline (8 workers) | 3.79 |
| Baseline (12 workers) | 4.42 |
| Baseline (16 workers) | 4.55 |

focus on gameplay interactions. Gameplay data offer behavioral diversity while minimizing privacy concerns, which enables broad community contribution and data sharing. Using the `ocap` desktop recorder for efficient collection, 14 human annotators recorded the dataset. The dataset comprises 335 hours of newly collected human demonstrations across 31 games. It spans diverse genres, including 3D third-person games such as GTA V and Cyberpunk 2077, first-person games like Apex Legends and Minecraft, and 2D top-down games like Brotato and Stardew Valley. This variety captures a wide range of visual environments and interaction styles, making it well-suited for vision-action pretraining. Further details on the dataset and collection process are provided in Appendix B.

## 4 GENERALIST INVERSE DYNAMICS MODEL

Collecting large-scale action data through manual demonstrations is infeasible due to prohibitive costs. The OWA Toolkit (Section 3) closes the instrumentation gap and standardizes over 2.6k hours of synchronized trajectories (Table 12), yet human capture alone remains a bottleneck relative to the ocean of unlabeled gameplay available online. VPT (Baker et al., 2022) addressed this by leveraging Inverse Dynamics Models (IDMs) to pseudo-label YouTube videos, but was limited to *Minecraft*, restricting generalization and dataset diversity. We train a **Generalist-IDM** on our multi-domain corpus collected via the OWA Toolkit, enabling generalization across heterogeneous interaction patterns. Our model can infer actions in out-of-distribution environments never seen during training, as demonstrated in Section 5.1. This capability enables pseudo-labeling of large-scale YouTube gameplay videos across diverse games, laying the foundation for internet-scale dataset collection.

### 4.1 TIMESTAMP-BASED EVENT TOKENIZATION

We represent desktop interactions as discrete *events*, each serialized into a short token sequence bounded by <EVENT_START> and <EVENT_END>. Observation events capture screen updates (*Screen Events*), while action events represent user inputs: *Keyboard Events* (key presses/releases) and *Mouse Events* (clicks, movements, scrolls). This event-level serialization unifies heterogeneous inputs into a consistent sequential representation for transformer modeling (Vaswani et al., 2017). For example, the tokens emitted for a single event follow the format below:

$$\text{<EVENT\_START>\{TYPE\}\{TIMESTAMP\}\{DETAIL\}</EVENT\_END>} \quad (1)$$

While most existing IDMs adopt a *tick-based prediction* (Baker et al., 2022; Ye et al., 2024)—predicting actions at fixed intervals—our design employs *timestamp-based prediction*. Unlike tick-based approaches that use a fixed prediction window (e.g., 50 ms), our IDM directly predicts both the event and its timestamp, preserving the asynchronous timing captured by `ocap` and converted corpora. This design provides two key advantages. First, it maintains cross-modal alignment with-

out resampling, allowing screen, keyboard, and mouse streams to stay synchronized even when their natural cadences differ. Second, timestamp-based prediction avoids generating empty ticks when no actions occur. By skipping unnecessary "no-op" tokens, our approach makes more efficient use of the limited inference context, enabling denser packing of relevant information and improving the efficiency of both learning and inference. A detailed specification of the event tokenization process is provided in Appendix C.

## 4.2 NEP-$\tau$: NEXT-EVENT PREDICTION WITH TEMPORAL OFFSET

Once raw desktop interactions are converted to event token sequences, we train the Generalist-IDM with a next-event-prediction objective. Given a trajectory consisting of observed states and actions $(o_1, a_1, o_2, a_2, \ldots, o_T)$, where each action $a_t$ is taken at state $o_t$ and leads to state $o_{t+1}$, the goal is to predict action $a_t$ based on all preceding observations and actions. This objective enables the model to learn mappings between observed states and actions while preserving temporal dependencies within the trajectory.

$$\mathcal{L}_{\text{NEP}} = -\mathbb{E}_{(o_{1:T}, a_{1:T}) \sim \mathcal{D}} \left[ \sum_{t=1}^{T} \log P_\theta \big( a_t \mid o_{1:t},\, a_{1:t-1} \big) \right] \tag{2}$$

Inspired by IDM-K (Tot et al., 2025), which conditions on extended future trajectories to improve inverse dynamics, we adopt NEP-$\tau$, a temporal-offset variant of NEP. Unlike IDM-K, which jointly encodes entire past and future trajectories, our method simply rearranges the (observation, action) sequences by shifting the observation window forward by $\tau$ steps. This allows the model to incorporate future observations up to $\tau$ steps ahead without encoding entire future trajectories, enhancing temporal consistency. Formally, the objective is:

$$\mathcal{L}_{\text{NEP-}\tau} = -\mathbb{E}_{(o_{1:T}, a_{1:T}) \sim \mathcal{D}} \left[ \sum_{t=1}^{T} \log P_\theta \Big( a_t \mid o_{1:\,\min(t+\tau, T)},\, a_{1:t-1} \Big) \right] \tag{3}$$

We ablate $\tau \in \{0, 50, 100, 150, 200\}$ ms across six in-distribution games (Table 18). Without any offset ($\tau = 0$), Pearson correlations collapse near zero and keyboard accuracy drops sharply, confirming that future context is essential for resolving the current action. A small offset ($\tau = 50$ ms) recovers mouse prediction but remains suboptimal for keyboard accuracy. Performance stabilizes at $\tau \geq 100$ ms with only minor variation up to 200 ms, showing that NEP-$\tau$ is robust to the exact offset once sufficient future context is provided. Based on these results, we adopt $\tau = 100$ ms as the default in all experiments.

| $\tau$ (ms) | Avg Pearson X | Avg Pearson Y | Avg Scale X | Avg Scale Y | Avg Keypress (Kbd) | Avg Keypress (Mouse) |
|---|---|---|---|---|---|---|
| 0 | 0.108 | 0.050 | 25.66 | 22.30 | 0.386 | 0.961 |
| 50 | 0.803 | 0.734 | 1.17 | 1.48 | 0.492 | 0.948 |
| 100 | 0.796 | 0.783 | 1.23 | 1.31 | 0.730 | 0.957 |
| 150 | 0.891 | 0.818 | 1.08 | 1.14 | 0.767 | 0.965 |
| 200 | 0.893 | 0.837 | 1.28 | 1.49 | 0.740 | 0.947 |

Table 3: Aggregate performance across all games for different temporal offsets $\tau$. Per-game breakdown is provided in Appendix G.1.

## 4.3 PSEUDO-LABELING WITH YOUTUBE GAMEPLAY VIDEOS

We focus on pseudo-labeling gameplay videos because they are abundant, actively shared, and largely free of personally identifiable content, sidestepping the privacy concerns. YouTube gameplay footage also exhibits consistent HUD layouts and frame rates, which align well with the OWA Toolkit's event schema. Our pipeline first curates long-form gameplay uploads with permissive licenses, retrieves them at 20 Hz, and converts the frames into *Screen* events so they can be fed through the same tokenizer used for human demonstrations. Building on this, we train the Generalist-IDM using the InternVL3-1B (Zhu et al., 2025) architecture with the NEP-$\tau$ objective. The Generalist-IDM then autoregressively produces *Keyboard* and *Mouse* events via the NEP-$\tau$ objective between consecutive screen frames, using a temporal stopping criterion and sliding context window (Al-

gorithm 1), after which we apply consistency checks before materializing the pseudo-labels (Appendix B).

Applying this procedure contributes 1055 hours of additional trajectories across twenty publicly shared titles, as summarized in Table 14, complementing the curated corpus described in Table 12 and Section 3. Importantly, because our model is designed to be *generalist*, we do not require any filtering of domain-specific interfaces such as inventory menus or map screens. Instead, these heterogeneous visual contexts are naturally included as part of the pseudo-labeled demonstrations, broadening the scope of training data without additional heuristics. These pseudo-labeled trajectories form the seed for scaling desktop vision-action pretraining to internet-scale data sources.

## 5 RESULTS

### 5.1 PERFORMANCE OF THE GENERALIST-IDM

**In-Distribution Performance.** We begin by evaluating the Generalist-IDM on six in-distribution video games spanning both 2D and 3D settings, comparing its performance to Specialist-IDMs trained individually on each game. We employ an autoregressive inference pipeline to generate actions and evaluate model performance across multiple metrics. Further details are provided in Appendix F. As shown in Table 4 and Table 5, our Generalist-IDM achieves strong performance across all environments. Notably, it yields large gains in Pearson correlation (e.g., +39.5 points on Stardew Valley X) and Keyboard accuracy (e.g., +57.6 points on Brotato), demonstrating robust generalization over diverse control dynamics.

| Game | Model | Pearson | | Scale Ratio | | Keypress Acc. | |
|------|-------|---------|---|-------------|---|---------------|---|
| | | X | Y | X | Y | Kbd | Mouse |
| Brotato | IDM | 65.92 | 67.56 | 1.04 | 1.04 | 28.80 | 97.59 |
| | G-IDM | 73.65 | 82.03 | 1.37 | 1.29 | 86.36 | 98.50 |
| Stardew Valley | IDM | 43.47 | 63.69 | 1.19 | 1.18 | 69.35 | 91.90 |
| | G-IDM | 82.98 | 75.57 | 1.13 | 1.17 | 74.35 | 96.43 |
| Core Keeper | IDM | 48.03 | 62.09 | 1.15 | 1.17 | 69.42 | 92.33 |
| | G-IDM | 77.25 | 64.55 | 1.43 | 1.51 | 70.00 | 94.01 |

Table 4: Evaluation results on 2D games

| Game | Model | Pearson | | Scale Ratio | | Keypress Acc. | |
|------|-------|---------|---|-------------|---|---------------|---|
| | | X | Y | X | Y | Kbd | Mouse |
| Apex Legends | IDM | 65.16 | 57.84 | 1.29 | 1.25 | 67.47 | 99.33 |
| | G-IDM | 83.90 | 85.27 | 1.13 | 1.23 | 76.55 | 99.67 |
| GTA V | IDM | 63.64 | 81.08 | 1.39 | 1.23 | 58.13 | 94.65 |
| | G-IDM | 79.44 | 83.89 | 1.09 | 1.42 | 69.83 | 94.11 |
| Minecraft | IDM | 59.83 | 63.83 | 1.20 | 1.22 | 53.54 | 82.48 |
| | G-IDM | 80.29 | 78.38 | 1.24 | 1.27 | 60.97 | 91.65 |

Table 5: Evaluation results on 3D games

**Out-of-Distribution Generalization.** We evaluate the generalization of our Generalist-IDM on two unseen games: Battlefield 6 (3D) and Ogu and the Secret Forest (2D). In Battlefield 6, the Generalist-IDM achieves $63\%$ keyboard accuracy, matching or slightly outperforming the Specialist-IDM, indicating solid transfer to an unseen FPS similar to the training set. Moreover, when provided with a few-shot prefix that fills the first 2048 tokens in our streaming inference, the predicted scale ratio improves significantly—indicating that the Generalist-IDM exhibits an in-context ability to adapt to mouse sensitivity. In Ogu and the Secret Forest, the Generalist-IDM more than doubles the Specialist-IDM's performance (from about $12\%$ to nearly $28\%$), showing substantial gains even under a large domain gap. Taken together, these results demonstrate that the Generalist-IDM is capable of adapting across both familiar and substantially different environments.

### 5.2 TRANSFERABILITY TO DOWNSTREAM TASKS

To validate the transfer of useful knowledge from the desktop domain to the embodied AI domain, we evaluate our D2E framework on both robot manipulation and navigation tasks. For manipulation, we first assess performance in simulated environments using the LIBERO (Liu et al., 2023) and Meta-World (Yu et al., 2020) benchmarks, and then further verify effectiveness in the real world by following the evaluation protocol used in SmolVLA (Shukor et al., 2025). For navigation, we evaluate in simulation using the CANVAS benchmark (Choi et al., 2024). Collectively, these experimental results demonstrate that our D2E framework effectively transfers knowledge across domains, resulting in strong performance on robotics downstream tasks.

For these experiments, we use the InternVL3-1B model as our backbone, which is also the architecture used in our Generalist-IDM. We train this model under two different settings: **VAPT without pseudo-labels** (259 hours), which uses only the human-collected dataset, and **VAPT with pseudo-labels** (1.3K hours), which augments the human data with a pseudo-labeled dataset generated from

| Model | Pearson | | Scale Ratio | | Keypress Acc. | |
|---|---|---|---|---|---|---|
| | X | Y | X | Y | Kbd | Mouse |
| *Battlefield 6* | | | | | | |
| IDM (FT) | 57.28 | 61.74 | 1.00 | 1.00 | 62.44 | 94.55 |
| G-IDM (ZS) | 57.36 | 63.17 | 3.13 | 3.56 | 47.75 | 92.11 |
| G-IDM (FS) | 56.79 | 63.40 | 1.07 | 1.05 | 52.64 | 93.89 |
| G-IDM (FT) | 54.90 | 62.89 | 1.06 | 1.04 | 58.55 | 93.41 |
| *Ogu Forest* | | | | | | |
| IDM (FT) | – | – | – | – | 11.73 | – |
| G-IDM (ZS) | – | – | – | – | 27.80 | – |
| G-IDM (FS) | – | – | – | – | 27.97 | – |
| G-IDM (FT) | – | – | – | – | 26.88 | – |

Table 6: Out-of-distribution performance on unseen 3D and 2D games. Note that *Ogu Forest* uses only keyboard inputs.

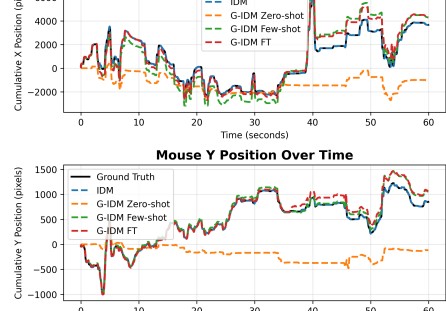

Figure 4: Trajectory of Battlefield 6.

YouTube videos using the Generalist-IDM. Further training details can be found in Appendix E, and detailed experimental setups for each downstream task are provided in Appendix H.

**Why Desktop-to-Embodied Transfer May Work.** We identify three properties of desktop pre-training that may facilitate transfer to embodied domains. First, *action modality alignment*: VAPT is trained on explicit vision–action trajectories rather than solely on image–text pairs, encouraging the model to internalize how visual observations correspond to motor commands. Second, *goal-directed sequential decision-making*: desktop gameplay requires visual grounding, temporal reasoning, and long-range dependency modeling—capabilities that translate directly into coherent robotic control. Third, *high diversity*: our 20-game corpus spans 2D and 3D environments with heterogeneous mechanics, encouraging general-purpose control priors instead of domain-specific shortcuts. Empirically, training loss curves (Appendix G.2) corroborate these hypotheses: VAPT-initialized models converge immediately, whereas baseline models exhibit an initial plateau, indicating that desktop pretraining provides better-aligned representations for embodied control.

**Robot Manipulation.** For manipulation, we first evaluate our VAPT models on the LIBERO benchmark (Liu et al., 2023). As shown in Table 7, the InternVL3-1B baseline performs relatively poorly. VAPT without pseudo-labels achieves a substantial improvement, reaching 96.6% on Total and 93.6% on long-horizon tasks. These results are comparable to or even surpass those of much larger models such as $\pi_0$ (3.3B) and OpenVLA (7B). Interestingly, incorporating pseudo-labels does not provide additional gains on manipulation tasks. We attribute this to the nature of manipulation tasks, where precise human supervision is more critical than data scale and diversity. Overall, our 1B-parameter model matches or outperforms significantly larger policies such as $\pi_0$ (3.3B) and OpenVLA (7B), with particularly strong advantages on long-horizon tasks that require careful action sequencing.

| Method | Params | VLA Pt | Spatial | Object | Goal | 10 (long) | Total |
|---|---|---|---|---|---|---|---|
| Octo (Octo Model Team et al., 2024) | 93M | Yes | 78.9 | 85.7 | 84.6 | 51.1 | 75.1 |
| OpenVLA (Kim et al., 2024) | 7B | Yes | 84.7 | 88.4 | 79.2 | 53.7 | 76.5 |
| DiT Policy (Dasari et al., 2025) | 115M | No | 84.2 | 96.3 | 85.4 | 63.8 | 82.4 |
| $\pi_0$ Black et al. (2024) | 3.3B | Yes | 90.0 | 86.0 | 95.0 | 73.0 | 86.0 |
| SmolVLA (Shukor et al., 2025) | 2.25B | No | 93.0 | 94.0 | 91.0 | 77.0 | 88.7 |
| PI-KI (Driess et al., 2025) | 300M | Yes | 98.0 | 97.8 | 95.6 | 85.8 | 94.3 |
| OpenVLA-OFT (Kim et al., 2025) | 7B | Yes | 97.6 | 98.4 | 97.9 | 94.5 | 97.1 |
| Baseline (InternVL3-1B) | 1B | No | 94.4 | 97.0 | 93.6 | 54.2 | 84.8 |
| + VAPT w/o pseudo | 1B | No | 95.8 | 98.4 | 98.6 | 93.6 | 96.6 |
| + VAPT w/ pseudo | 1B | No | 89.6 | 98.2 | 93.8 | 87.2 | 92.2 |

Table 7: Evaluation results on Libero benchmark (success rates, %).

Next, we evaluate our VAPT models on Meta-World (Yu et al., 2020), a standard benchmark for multi-task robotic manipulation. We compare VAPT against the InternVL3-1B baseline across tasks

of varying difficulty. Even without robotics-specific pretraining or extensive hyperparameter tuning, VAPT consistently outperforms the baseline, showing an average success rate improvement of roughly 5% (a ~25% relative gain). The performance gap is most pronounced in the *Hard* and *Very Hard* categories (e.g., 8.0% vs. 20.0–24.0% on *Very Hard*), suggesting that the priors learned from desktop data are particularly robust for complex manipulation challenges.

We further validate our approach with a real-world pick-and-place experiment using an SO101 robot arm, following the evaluation protocol of SmolVLA (Shukor et al., 2025). The task requires grasping a blue cube and placing it in a white box, with the cube placed at five distinct initial positions. We collect 208 demonstration episodes and evaluate each trained policy over 30 rollouts (further details in Appendix H). As shown in Table 8, the baseline InternVL3-1B achieves 70% success rate, while both VAPT variants reach 80%, confirming that VAPT transfers effectively to real-world hardware.

| Method | Meta-World | | | | | SO101 |
|---|---|---|---|---|---|---|
| | Easy | Medium | Hard | Very Hard | Avg | Pick&Place |
| Baseline (InternVL3-1B) | 55.4 | 14.5 | 1.7 | 8.0 | 19.9 | 70.0 |
| + VAPT w/o pseudo | 53.6 | 18.2 | 8.3 | 20.0 | **25.0** | **80.0** |
| + VAPT w/ pseudo | 52.1 | 16.4 | 6.7 | 24.0 | 24.8 | **80.0** |

Table 8: Evaluation results on Meta-World and real-world SO101 pick-and-place (success rates, %).

**Robot Navigation.** For robot navigation, we evaluate on the CANVAS (Choi et al., 2024) benchmark, which tests robustness to both misleading and precise instructions across diverse simulated environments. Compared to the baseline, our VAPT framework shows clear gains: without pseudo-labels, performance matches the baseline (75.3%), while adding pseudo-labeled demonstrations increases performance to 83.3%, an 8-point improvement. The benefit is especially large under misleading instructions, as in *sim_orchard* (86.7% vs. 53.3%) and *sim_street_sidewalk* (73.3% vs. 40.0%), whereas performance under precise instructions remains near ceiling. These results indicate that pseudo-labeling is particularly useful for navigation tasks, where success depends on high-level planning rather than precise low-level control.

| Method | Gallery | | Office | | Orchard | | Street (road) | | Street (side) | | Total |
|---|---|---|---|---|---|---|---|---|---|---|---|
| | Mis. | Prec. | Mis. | Prec. | Mis. | Prec. | Mis. | Prec. | Mis. | Prec. | |
| Baseline (InternVL3-1B) | **53.3** | **100.0** | **100.0** | **100.0** | 53.3 | 40.0 | **94.4** | **100.0** | 40.0 | 73.3 | 75.3 |
| + VAPT w/o pseudo | 33.3 | 93.3 | 93.3 | **100.0** | 53.3 | 53.3 | 88.9 | 91.7 | 53.3 | **93.3** | 75.3 |
| + VAPT w/ pseudo | **53.3** | 93.3 | **100.0** | **100.0** | 86.7 | 60.0 | 88.9 | **100.0** | **73.3** | 80.0 | **83.3** |

Table 9: Evaluation results on CANVAS navigation benchmark (success rates, %). Mis. = misleading instructions, Prec. = precise instructions.

# 6 CONCLUSION

Embodied AI has long struggled with the prohibitive cost of collecting large-scale physical interaction data, limiting its ability to benefit from internet-scale resources. To address this challenge, we proposed using desktop interactions as an abundant and low-cost substrate for pretraining. Our contributions are threefold: (1) the OWA Toolkit, which standardizes and compresses diverse desktop data into a scalable format; (2) the Generalist-IDM, a timestamp-based inverse dynamics model that generalizes across unseen games and demonstrates a pathway toward internet-scale pseudo-labeling; and (3) VAPT, which explores the transfer of desktop-pretrained representations to robotics tasks. Leveraging 1.3K+ hours of human and pseudo-labeled data, our framework achieves 96.6% success on LIBERO manipulation and 83.3% on CANVAS navigation, demonstrating that digital sensorimotor patterns can directly improve embodied AI benchmarks. We release all our tools, datasets, and models publicly to enable the community to build upon this foundation and further investigate desktop-to-embodied transfer. These results establish desktop data as a practical and scalable resource for advancing embodied intelligence, opening a new path toward general-purpose agents without relying on prohibitively expensive physical data collection.

## REPRODUCIBILITY STATEMENT

To ensure full reproducibility of our work, we release comprehensive resources and documentation. All source code for the OWA Toolkit (`ocap` recorder and OWAMcap format implementation), Generalist-IDM training, and downstream task fine-tuning is publicly available at `https://github.com/worv-ai/D2E`, including detailed installation instructions and usage examples. The complete 2.6K hour desktop dataset (335 hours newly collected, 2.3K hours converted) and 1K+ hours of pseudo-labeled data are accessible through the same repository with standardized OWAMcap format specifications described in Section 3 and Appendix A. Pre-trained model weights for both Generalist-IDM and VAPT foundation models are provided along with training configurations. Hyperparameters and training schedules are detailed in Appendix E, including batch sizes, learning rates, and hardware requirements (8 H100 GPUs for IDM training). Data preprocessing pipelines, including temporal offset implementation (Section 4) and event tokenization schemes (Appendix C), are fully documented with reference implementations. Evaluation protocols and metrics are specified in Appendix F with corresponding evaluation scripts in the repository. For compute-constrained researchers, we release smaller dataset subsets and checkpoint models at various training stages to facilitate partial reproduction and ablation studies.

## ACKNOWLEDGMENTS

This work was partly supported by an Institute of Information & communications Technology Planning & Evaluation (IITP) grant funded by the Korean Government (MSIT) (No. RS-2021-II211343, Artificial Intelligence Graduate School Program (Seoul National University)), the Technology Innovation Program(RS-2025-25456760, Development of a humanoid robot specialized in chemical processes based on AI foundation model) funded by the Ministry of Trade, Industry and Resources (MOTIR, Korea).

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

# A OWA TOOLKIT DETAILS

## A.1 FORMAT COMPARISON

Prior desktop datasets commonly adopt one of two storage strategies. The LeRobot dataset (Cadene et al., 2024), CS:GO dataset (Pearce & Zhu, 2022), and the CraftJarvis "minecraft-vla-sft" dataset (He et al., 2025) store image-encoded frames directly in a single, monolithic table. While this layout is sufficient for training, it is ill-suited for recording because long-table stores typically do not support efficient real-time appends. By contrast, the VPT dataset (Baker et al., 2022) packages each sample as an MP4–JSONL pair. However, JSONL lacks the ability to interleave heterogeneous, typed streams with chunking and indexing. In practice, this limitation results in poor or unavailable topic-wise random seeking and reduced crash-safety, as writes are unreliable under unexpected termination. Furthermore, datasets that rely on image encoding are substantially less storage-efficient compared to standard video codecs.

The robotics community has encountered similar multimodal logging challenges. Traditional ROS bags exhibit performance and extensibility limitations (Foxglove, 2021), which motivated the development of the MCAP format (Foxglove, 2022): an open-source container format designed with efficient indexing and compression. MCAP has since become the de facto logging standard for ROS 2 (Foxglove, 2022; Foxglove Developers, 2024), demonstrating the benefits of specialized data formats for embodied AI research. However, no equivalent standard has been established for desktop datasets, motivating our introduction of the OWAMcap format.

## A.2 COMPRESSION EFFICIENCY

OWAMcap achieves substantial storage savings across multiple datasets, demonstrating its efficiency and scalability. For the CS:GO dataset (Pearce & Zhu, 2022), replacing the original HDF5 storage with OWAMcap (mkv+mcap) reduces the storage requirement from 689 GiB to 20 GiB—a $34.45\times$ reduction. Similarly, converting the VPT dataset (Baker et al., 2022) from JSONL to OWAMcap (mcap format) shrinks disk usage from 1.06 TiB to 7.12 GiB, achieving a $152\times$ reduction. This significant compression arises from two different aspects: (1) from using video encoding instead of saving raw image buffer on the CS:GO dataset's HDF5 and (2) from mcap's efficiency in representing/storing information on the VPT dataset's jsonl.

## A.3 VIDEO COMPRESSION PERFORMANCE

Another advantage of OWAMcap is MediaRef, a flexible system supporting storing media on (1) embedded or (2) external media. We support storing media in both external image files and external video files. This flexible design provides the opportunity to acquire significant compression efficiency through video encoding, such as H.265/HEVC. To further evaluate the benefits of video encoding, we benchmarked video compression performance for various encodings. Table 10 shows that video encoding provides superior compression rates while maintaining visual quality, enabling large-scale storage without compromising data fidelity. `ocap` is storing all media in H.265 by default and we observed similar compression ratio for recorded files.

| Format | Size per Frame | Total Size | Compression Ratio |
|---|---|---|---|
| Raw BGRA | 5.97 MB | 4.2 GB | 1.0× (baseline) |
| PNG | 1.87 MB | 1.31 GB | 3.2× |
| JPEG (Quality 85) | 191 KB | 135 MB | 31.9× |
| H.265 (keyframe 0.5s) | 27.8 KB | 19.6 MB | 217.8× |

Table 10: **Compression performance comparison for various encoding on our recorded Minecraft video.** Desktop screen capture at 1920×1080 resolution, 12 seconds @ 60 Hz. H.265 encoding uses nvd3d11h265enc for hardware acceleration. Video encoding yields significantly higher compression ratios than other formats. `ocap` is storing all media in H.265 by default and we observed similar compression ratio for recorded files. Note that size per frame for H.265 is an average over all frames, as keyframes are larger.

### A.4 OCAP ARCHITECTURE

The implementation of ocap is designed to maximize recording performance and reliability. ocap leverages Windows APIs, including DXGI (Microsoft Corporation) for hardware-accelerated screen capture, WASAPI for low-latency audio recording, and direct input event capture for precise keyboard and mouse logging. The media pipeline is built on GStreamer (GStreamer Team) and employs H.265/HEVC encoding (ITU-T, 2024; Sullivan et al., 2012) to achieve high compression efficiency while maintaining visual quality. The overall architecture, shown in Figure 5, integrates video, audio, and interaction streams within the OWAMcap format while ensuring synchronized, crash-safe recording.

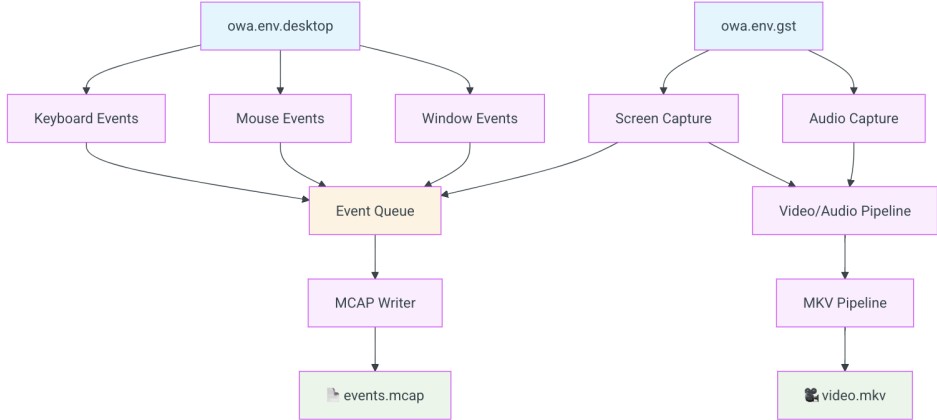

Figure 5: Architecture of ocap desktop recorder.

### A.5 SCREEN CAPTURE PERFORMANCE BENCHMARKS

ocap employs H.265/HEVC encoding for video content and AAC encoding for audio streams, enabling real-time recording with minimal system overhead. Table 11 compares the capture performance of ocap against existing alternatives, showing that our implementation consistently achieves higher frame rates and lower CPU utilization while preserving recording fidelity.

| Library | Avg. Time per Frame | Relative Speed |
|---|---|---|
| owa.env.gst | 5.7 ms | 1.0× (baseline) |
| pyscreenshot | 33 ms | 5.8× slower |
| PIL | 34 ms | 6.0× slower |
| MSS | 37 ms | 6.5× slower |
| PyQt5 | 137 ms | 24× slower |

Table 11: **Screen capture performance comparison.** Benchmarked on Intel i5-11400 with GTX 1650. ocap achieves 6× faster performance than common alternatives through Windows API and GStreamer integration.

### A.6 COMPARISON WITH EXISTING RECORDERS

To assess feature coverage and efficiency, we compared ocap against commonly used desktop recording frameworks. As shown in Figure 6, ocap is the only system that provides synchronized multimodal recording, robust crash-safety guarantees, and efficient compression in a single framework. These advantages make ocap a uniquely comprehensive solution for large-scale desktop interaction logging.

| Feature | ocap | OBS | wcap | pillow/mss |
|---|---|---|---|---|
| Advanced data formats (MCAP/MKV) | ✅ Yes | ❌ No | ❌ No | ❌ No |
| Timestamp aligned logging | ✅ Yes | ❌ No | ❌ No | ❌ No |
| Customizable event definition & Listener | ✅ Yes | ❌ No | ❌ No | ❌ No |
| Single python file | ✅ Yes | ❌ No | ❌ No | ❌ No |
| Audio + Window + Keyboard + Mouse | ✅ Yes | ⚠️ Partial | ❌ No | ❌ No |
| Hardware-accelerated encoder | ✅ Yes | ✅ Yes | ✅ Yes | ❌ No |
| Supports latest Windows APIs | ✅ Yes | ✅ Yes | ✅ Yes | ❌ No (legacy APIs only) |
| Optional mouse cursor capture | ✅ Yes | ✅ Yes | ✅ Yes | ❌ No |

Figure 6: Comparison of key features between `ocap` and other desktop recording tools.

### A.7 DATA PIPELINE OPTIMIZATIONS

Our data pipeline incorporates several key optimizations to address the limitations of conventional video processing approaches for foundation model training.

**Baseline Video Properties** To understand the limitations of default video encoding, we analyzed a representative sample from our dataset. The baseline configuration uses default x264 parameters, resulting in variable GOP structure that impacts random access performance. Frame type distribution shows: I-frames 57 (0.9%), B-frames 3847 (64.1%), P-frames 2096 (34.9%). I-frame interval analysis reveals significant variability: minimum 1.35s, maximum 12.50s, average 5.32s, median 4.83s. This variable GOP size creates inconsistent seeking performance, motivating our optimized x264 parameters.

**Optimized x264 Parameters** Default video encoding with x264 creates variable GOP structures with unpredictable keyframe intervals, causing inconsistent random access performance during training. Our optimization fixes keyframe intervals to 30 frames (1.5 seconds at 20 Hz) and disables B-frames entirely. This creates predictable GOP structure: I-P-P-P-...-P-I, enabling consistent random access performance. The elimination of B-frames reduces decoding complexity during seeking operations, while fixed keyframe intervals ensure uniform seeking distances.

**FSLDataset Construction** FSLDataset preserves episode temporal structure during sequence packing. For each episode, we sequentially list all events (screen, keyboard, mouse) in chronological order, then concatenate episodes sequentially until reaching the maximum sequence length (e.g., 4096 tokens). When an episode completes before reaching the maximum length, packing terminates immediately and remaining positions are padded. This approach maintains episode coherence while enabling uniform sequence lengths for efficient batch processing.

**Adaptive Batch Decoding Strategy** The baseline configuration uses single-frame decoding where each frame within an FSLDataset sample requires individual video seek and decode operations. For an FSLDataset sample containing $n$ frames, the baseline performs $n$ separate video decoder calls,

each involving: (1) seeking to the target frame position, (2) decoding from the nearest keyframe to the target frame, and (3) extracting the single target frame. This approach results in significant redundant I/O operations when multiple frames from the same video segment are needed.

Our adaptive batch decoding strategy processes all $n$ frames within each FSLDataset sample through a single batched operation, eliminating redundant seeking and keyframe decoding overhead. Both TorchCodec (PyTorch Team, 2024) v0.6.0 and our implementation use this per-sample batching approach: for each FSLDataset sample, we issue a single batched query that requests all images within the sample at once (no cross-sample batching or parallel workers).

## A.8 Benchmark Configuration

To quantify the effect of our optimized pipeline, we conduct comprehensive benchmarks across different configurations and training scenarios.

**Media Decoding Benchmark Setup**   The media decoding benchmark uses a representative FSL-Dataset containing 64 episodes of 5-minute Minecraft gameplay recorded at 640×360 resolution and 20 Hz frame rate. The FSLDataset is configured with fixed sequence length of 4096 tokens, where all sequences are tokenized and packed to this uniform length.

We measure performance using single-worker random-access iteration and report: (i) image throughput (img/s) calculated by dividing the total number of images by the time required to process all images during decoding, and (ii) average disk bytes read per image (KB/img) obtained by monitoring total bytes read during iteration divided by the number of images, capturing seeking and GOP decode overhead.

For I/O efficiency measurement, we create an isolated temporary filesystem and store all media data referenced by the FSLDataset in this dedicated path. During benchmarking, we monitor the total amount of data read from this filesystem to obtain precise I/O measurements.

Progressive configurations test: (1) baseline with default x264 parameters and single-frame decoding, (2) baseline + optimized x264 parameters, (3) optimized x264 + TorchCodec v0.6.0 batch decoding, and (4) optimized x264 + our adaptive batch decoding strategy. All benchmark experiments were conducted three times to ensure result stability, with all runs showing consistent performance within measurement variance. The reported results represent the final experimental run.

**InternVL3-1B Training Configuration**   Model training benchmarks use single H100 GPU with batch size 1, DeepSpeed Zero1 for memory optimization, FlashAttention 3 for efficient attention computation, and context length 4096 tokens. The baseline configuration uses default x264 parameters without batch decoding, while our optimized pipeline combines optimized x264 parameters with adaptive batch decoding API. We measure training throughput (iterations per second) across different numbers of dataloader workers to evaluate scalability and efficiency gains.

## B Dataset Details

### B.1 Collection and Quality Assurance

We collected the dataset using a distributed approach supported by contributions from community volunteers. To ensure participant privacy, we applied automated detection techniques followed by manual review to remove any sensitive information. Quality assurance involved both automated and manual procedures. Automated validation checked for temporal alignment issues and corrupted recordings, while human annotators manually evaluated the realism and fidelity of recorded behaviors. The final dataset captures a wide range of desktop interaction patterns, including navigation behaviors, application switching, text input, menu interactions, and multi-step task execution.

### B.2 Annotator Calibration and Protocols

Before recording, contributors completed an `ocap` calibration wizard that verified refresh rate, display resolution, cursor fidelity, and input-device mapping. Annotators—either modestly compensated participants or volunteers—followed standardized game prompts covering navigation, combat,

| Game/Application | Category | Genre | External | Hours |
|---|---|---|---|---|
| Apex Legends | ID | FPS | No | 25.8 |
| Euro Truck Simulator 2 | ID | Driving | No | 19.7 |
| Stardew Valley | ID | Top-Down Sim | No | 16.1 |
| Cyberpunk 2077 | ID | Open-World, RPG | No | 14.6 |
| Rainbow Six Siege | ID | FPS | No | 13.8 |
| Grand Theft Auto V | ID | Open-World, Driving | No | 11.7 |
| Slime Rancher | ID | Simulation | No | 11.1 |
| Medieval Dynasty | ID | Simulation, RPG | No | 10.7 |
| Dinkum | ID | Sandbox, Survival | No | 10.5 |
| Raft | ID | Survival, Co-op | No | 10.3 |
| Satisfactory | ID | Factory-Building | No | 10.1 |
| Minecraft (SP 1.21.8) | ID | Open-World, Sandbox | No | 10.1 |
| Grounded | ID | Survival, Co-op | No | 10.1 |
| Ready Or Not | ID | Tactical FPS | No | 10.0 |
| Counter-Strike 2 | ID | FPS | No | 9.9 |
| Core Keeper | ID | Sandbox, Survival | No | 9.4 |
| Barony | ID | Roguelike RPG | No | 9.3 |
| Monster Hunter Wilds | ID | Action RPG | No | 8.7 |
| Brotato | ID | Top-Down Shooter | No | 6.1 |
| PUBG: Battlegrounds | ID | FPS, Battle Royale | No | 4.9 |
| **Total Used for Train and test** | | | | **258.7** |
| Ogu and the Secret Forest | OOD | Adventure, Puzzle | No | 2.3 |
| Battlefield 6 (Open Beta) | OOD | FPS | No | 2.3 |
| Eternal Return | Collection | MOBA, Survival | No | 17.3 |
| MapleStory Worlds-Southperry (EA) | Collection | Open-World, Sandbox | No | 14.1 |
| Overwatch | Collection | FPS, Hero Shooter | No | 10.3 |
| Enshrouded | Collection | Survival, RPG | No | 10.1 |
| Vampire Survivors | Collection | Top-Down Platformer | No | 2.8 |
| Skul | Collection | Roguelike Platformer | No | 2.0 |
| PEAK | Collection | Casual/Arcade | No | 1.8 |
| Super Bunny Man | Collection | Platformer, Co-op | No | 0.7 |
| VALORANT | Collection | FPS | No | 0.3 |
| **Total (Collected)** | | | | **335.6** |

Table 12: **Collected desktop data statistics.** The dataset includes internally collected demonstrations across diverse games and applications.

and resource-management scenarios; detailed environment statistics are listed in Table 12. All sessions were screen-captured at FHD or QHD 60 Hz with synchronized mouse and keyboard traces, and `ocap`'s turnkey workflow meant anyone could gather synchronized data with minimal setup; annotators re-ran the calibration sequence whenever their hardware changed.

| Game/Application | Category | Genre | External | Hours |
|---|---|---|---|---|
| Minecraft - VPT (Baker et al., 2022) | Converted | Open-World, Sandbox | Yes | 2194 |
| CSGO - CS_DM (Pearce & Zhu, 2022) | Converted | FPS | Yes | 100 |
| **Total (Converted)** | | | | **2294.0** |

Table 13: **Converted dataset statistics.** Converted data from existing public benchmarks complement the collected corpus.

## B.3 CONVERTED DATA

The converted dataset includes Minecraft demonstrations from Baker et al. (Baker et al., 2022) and Counter-Strike 2 data from Pearce et al. (Pearce & Zhu, 2022). These external sources were standardized into the OWAMcap format, ensuring consistency and seamless integration across different datasets.

## B.4 PREPROCESSED DATASET

Before training, we applied preprocessing to handle temporal offsets. Specifically, after applying a temporal offset $\tau$, only the sequences of action labels were shifted, while the observations remained unchanged. We use a temporal offset of $\tau = 100$ ms to preprocess the training data for both the generalist and specialist IDM models. Additionally, we filtered out inactive segments where no actions occurred for extended periods to reduce noise and improve training efficiency.

## B.5 PSEUDO-LABELED DATASET

We collect high-quality YouTube gameplay videos through a combination of targeted search and bulk download. For the search phase, we used the query template *"GAME_NAME no commentary,"* where the term *no commentary* is widely understood to indicate pure gameplay videos without additional overlays, commentary, or editing. After obtaining video links, we downloaded the videos using the open-source tool `yt-dlp`. To ensure consistency, we restricted the maximum resolution to 480p. In addition, frequent cookie renewal and a download rate cap of 62.5 Mb/s were necessary to bypass YouTube's automated bot detection mechanisms. Through this pipeline, we successfully curated over 1,000 hours of high-quality gameplay footage for pseudo-labeling. The total collected video duration per game is summarized in Table 14.

| Game | Duration (h) |
|---|---|
| Stardew Valley | 69.7 |
| Minecraft | 62.8 |
| Monster Hunter Wilds | 63.3 |
| Dinkum | 60.8 |
| Satisfactory | 59.8 |
| Cyberpunk 2077 | 58.5 |
| Medieval Dynasty | 58.4 |
| Raft | 58.0 |
| Core Keeper | 58.0 |
| Euro Truck Simulator 2 | 57.3 |
| Grounded | 57.2 |
| Rainbow Six | 56.3 |
| GTA 5 | 54.1 |
| Brotato | 52.6 |
| PUBG | 50.7 |
| Counter-Strike 2 | 49.8 |
| Apex Legends | 48.7 |
| Slime Rancher | 33.3 |
| Ready or Not | 29.0 |
| Barony | 16.7 |
| **Total** | **1054.8** |

Table 14: **Pseudo-labeled Duration by Game (G-IDM).** Total effective hours of successfully processed pseudo-labeled data per game.

## C  EVENT TOKENIZATION DETAILS

To train the Generalist IDM effectively, raw desktop interaction logs must be converted into a structured representation that the model can understand. We represent the entire interaction sequence as a stream of discrete *event tokens*. Each event corresponds to either an observation or an action. Observation events capture changes in the visual state of the environment, such as screen updates (*Screen Events*), while action events represent user inputs, including *Keyboard Events* (key presses and releases) and *Mouse Events* (clicks, movements, and scrolls).

By tokenizing data at the event level, we unify heterogeneous inputs into a consistent, sequential representation that can be modeled effectively using a single decoder-only transformer. This representation accommodates both asynchronous observations and actions while preserving fine-grained temporal alignment between them.

### C.1  EVENT TOKEN

We append specialized tokens to the model's vocabulary for desktop interaction modeling. **Event structure tokens** (<EVENT_START> and <EVENT_END>) delineate the boundaries of interaction sequences, while **event type tokens** (<KEYBOARD>, <MOUSE>, <SCREEN>) semantically categorize the modality of each event.

**Numeric encoding tokens** (<0> to <9>) serve multiple purposes:

- Mouse movement deltas are encoded using a configurable base system (default: [2, 10, 10, 10]), allowing efficient representation of signed values within a $\pm 1999$ pixel range.
- Mouse scroll values are similarly quantized using base-10 tokens.
- Timestamps are encoded using temporal bases (default: [10, 10, 10]), covering a 10-second window with 10ms resolution. Timestamps are cyclic, wrapping from 999 back to 000.

**Mouse interaction tokens** include:

- Sign tokens (<SIGN_PLUS>, <SIGN_MINUS>) for indicating the direction of movement deltas,
- Mouse button tokens (<MB_0> to <MB_15>) for encoding mouse button flags in hexadecimal.

**Keyboard interaction tokens** consist of:

- Virtual key code tokens (<VK_0> to <VK_255>) to represent all Windows virtual key inputs,
- Action tokens (<press>, <release>) to indicate key state transitions.

This factorized token design creates modular, modality-specific token spaces while maintaining a compact vocabulary. Mouse button flag definitions are provided in Table 15, and the full virtual key code mapping is shown in Table 16.

| Flag Name | Hex Value | Description |
|---|---|---|
| RI_MOUSE_NOP | 0x0000 | No operation |
| RI_MOUSE_LEFT_BUTTON_DOWN/UP | 0x0001/0x0002 | Left button press/release |
| RI_MOUSE_RIGHT_BUTTON_DOWN/UP | 0x0004/0x0008 | Right button press/release |
| RI_MOUSE_MIDDLE_BUTTON_DOWN/UP | 0x0010/0x0020 | Middle button press/release |
| RI_MOUSE_BUTTON_4_DOWN/UP | 0x0040/0x0080 | Side button 4 press/release |
| RI_MOUSE_BUTTON_5_DOWN/UP | 0x0100/0x0200 | Side button 5 press/release |
| RI_MOUSE_WHEEL | 0x0400 | Vertical scroll wheel |
| RI_MOUSE_HWHEEL | 0x0800 | Horizontal scroll wheel |

Table 15: Windows Raw Mouse Button Flags

| Key Name | VK Code | Description | Key Name | VK Code | Description |
|---|---|---|---|---|---|
| LBUTTON | 1 | Left mouse button | KEY_0–KEY_9 | 48–57 | '0'–'9' keys |
| RBUTTON | 2 | Right mouse button | KEY_A–KEY_Z | 65–90 | 'A'–'Z' keys |
| CANCEL | 3 | Control-break | LWIN | 91 | Left Windows key |
| MBUTTON | 4 | Middle mouse button | RWIN | 92 | Right Windows key |
| XBUTTON1 | 5 | X1 mouse button | APPS | 93 | Applications key |
| XBUTTON2 | 6 | X2 mouse button | NUMPAD0–9 | 96–105 | Numpad 0–9 |
| BACK | 8 | Backspace key | MULTIPLY | 106 | Numpad * |
| TAB | 9 | Tab key | ADD | 107 | Numpad + |
| CLEAR | 12 | Clear key | SUBTRACT | 109 | Numpad - |
| RETURN | 13 | Enter key | DECIMAL | 110 | Numpad . |
| SHIFT | 16 | Shift key | DIVIDE | 111 | Numpad / |
| CONTROL | 17 | Ctrl key | F1–F12 | 112–123 | F1–F12 function keys |
| MENU | 18 | Alt key | NUMLOCK | 144 | Num Lock |
| PAUSE | 19 | Pause key | SCROLL | 145 | Scroll Lock |
| CAPITAL | 20 | Caps Lock | LSHIFT | 160 | Left Shift |
| ESCAPE | 27 | Esc key | RSHIFT | 161 | Right Shift |
| SPACE | 32 | Spacebar | LCONTROL | 162 | Left Ctrl |
| PRIOR | 33 | Page Up | RCONTROL | 163 | Right Ctrl |
| NEXT | 34 | Page Down | LMENU | 164 | Left Alt |
| END | 35 | End key | RMENU | 165 | Right Alt |
| HOME | 36 | Home key | OEM_1 | 186 | ; : key |
| LEFT | 37 | Left arrow | OEM_PLUS | 187 | = + key |
| UP | 38 | Up arrow | OEM_COMMA | 188 | , < key |
| RIGHT | 39 | Right arrow | OEM_MINUS | 189 | - _ key |
| DOWN | 40 | Down arrow | OEM_PERIOD | 190 | . > key |
| INSERT | 45 | Insert key | OEM_2 | 191 | / ? key |
| DELETE | 46 | Delete key | OEM_3 | 192 | ' key |

Table 16: Windows Virtual Key Codes

## C.2 EVENT TOKEN STRUCTURE

All event tokens follow a consistent structure:

$$\texttt{<EVENT\_START>} <\texttt{event\_type}> <\texttt{timestamp}> <\texttt{event\_detail}> \texttt{<EVENT\_END>}$$

where:

- `<EVENT_START>` and `<EVENT_END>` are special tokens that delimit each event

- `<timestamp>` encodes the precise timing of the event in nanoseconds

- `<event_type>` specifies the type of event (e.g., `<SCREEN>`, `<KEYBOARD>`, `<MOUSE>`)

- `<event_detail>` contains event-specific information

## C.3 SCREEN EVENTS

Screen events capture visual observations from the desktop environment. Each screen event contains an image token sequence:

$$\texttt{<EVENT\_START><SCREEN>} <\texttt{timestamp}> <\texttt{image\_tokens}> \texttt{<EVENT\_END>}$$

For example:

$$\texttt{<EVENT\_START><SCREEN><1><8><5>}^{256}\texttt{<EVENT\_END>}$$

The timestamp `<1><8><5>` represents 185 time units, and the image is encoded using 256 visual tokens following the InternVL3 tokenization scheme.

## C.4 Keyboard Events

Keyboard events encode key press and release actions using virtual key code tokens:

<EVENT_START><KEYBOARD> $<$ timestamp $><$ vk_token $><$ action $>$ <EVENT_END>

For example:

<EVENT_START><KEYBOARD><2><0><0><VK_32><release><EVENT_END>

This represents a key release event at timestamp 200, where <VK_32> corresponds to the spacebar. The action can be either <press> or <release>.

## C.5 Mouse Events

Mouse events are the most complex among input modalities, as they encode continuous movement, discrete button states, and scroll actions.

<EVENT_START><MOUSE><timestamp><dx_sign><dx_magnitude><dy_sign> <dy_magnitude><button_flags><scroll_data><EVENT_END>

The optional <scroll_data> token is included only when the <button_flags> field indicates the presence of scroll wheel activity.

**Mouse Movement Example.** Consider the following mouse event:
<EVENT_START><MOUSE><2><4><5><SIGN_PLUS><0><0><0><2><SIGN_MINUS> <0><0><1><9><MB_4><MB_8><MB_0><SIGN_PLUS><0><EVENT_END>

This token sequence is decoded as follows:

**Timestamp:** <2><4><5> represents $2 \times 100 + 4 \times 10 + 5 = 245$ time units.

**Mouse Displacement:** The displacement uses separate sign and magnitude encoding:

dx: <SIGN_PLUS><0><0><0><2> $= +(0 \times 1000 + 0 \times 100 + 0 \times 10 + 2) + 2$ pixels  (4)

dy: <SIGN_MINUS><0><0><1><9> $= -(0 \times 1000 + 0 \times 100 + 1 \times 10 + 9) = -19$ pixels (5)

**Button Flags:** <MB_4><MB_8><MB_0> encodes button states as hexadecimal digits: $0x480_{16} = 1152_{10}$.

This corresponds to:

- 0x400: Vertical scroll wheel event
- 0x080: Mouse button 4 (side button) released

**Scroll Data:** <SIGN_PLUS><0> indicates no scroll delta (magnitude 0).

**Final Interpretation:** Mouse moved $dx = +2$, $dy = -19$ pixels at timestamp 245, with scroll wheel activity and side button release.

# D Model Architecture Details

For Generalist-IDM, we adopt the InternVL3-1B model (Zhu et al., 2025), which integrates InternViT as the vision encoder and Qwen2.5 (Yang et al., 2024) as the language backbone. InternVL3 is trained with native multimodal pretraining and demonstrates strong performance on video–text interleaved tasks, making it a suitable foundation for our work.

We expand the model's tokenizer by adding additional event tokens to represent events in our desktop data. Furthermore, we transfer the semantic initialization from corresponding regular language tokens to the newly added event tokens.

---

**Algorithm 1** Streaming Autoregressive Inference for Generalist-IDM

---

**Require:** Screen events $\mathcal{O} = \{o_1, o_2, \ldots, o_T\}$, model $P_\theta$, max context length $L$
**Ensure:** Predicted action events $\mathcal{A}$
 1: $\mathcal{C} \leftarrow \{o_1\}$ {Initialize context window}
 2: $\mathcal{A} \leftarrow \emptyset$
 3: **for** $t = 1$ **to** $T - 1$ **do**
 4:   **loop**
 5:     $\hat{a} \leftarrow \text{Generate}(P_\theta, \mathcal{C})$ {Predict next event via NEP-$\tau$}
 6:     **if** $\text{Decode}(\hat{a})$ fails **then**
 7:       **break** {Invalid event}
 8:     **end if**
 9:     **if** $\text{timestamp}(\hat{a}) > \text{timestamp}(o_{t+1})$ **then**
10:       **break** {Temporal stopping criterion}
11:     **end if**
12:     $\mathcal{C} \leftarrow \mathcal{C} \cup \{\hat{a}\}; \quad \mathcal{A} \leftarrow \mathcal{A} \cup \{\hat{a}\}$
13:   **end loop**
14:   $\mathcal{C} \leftarrow \mathcal{C} \cup \{o_{t+1}\}$ {Append next observation}
15:   **while** $|\mathcal{C}| > L$ **do**
16:     Remove oldest event from $\mathcal{C}$ {Trim context}
17:   **end while**
18: **end for**
19: **return** $\mathcal{A}$

---

## E  TRAINING DETAILS

The Generalist-IDM was trained on 8 H100 GPUs (80GB) for approximately 24 hours, totaling 192 H100-hours. The entire training process incurred a cost of only $\sim$ \$800 for training on 259 hours of human-collected data, highlighting the efficiency enabled by our OWA Toolkit.

All models were trained using a maximum context length of 8192 tokens. For the IDM models, both the generalist and specialist variants, we apply a temporal offset of $\tau = 100$ ms when constructing the training dataset.

We used the following training schedules:

- **Generalist-IDM**: 5 epochs
- **Specialist-IDM**: 5 epochs
- **Generalist-IDM (fine-tuning)**: 3 epochs
- **VAPT (w/o pseudo)**: 3 epochs on the human-collected vision-action dataset
- **VAPT (w/ pseudo)**: 1 epoch on the pseudo-labeled dataset, followed by 3 epochs on the human-collected dataset

All experiments were conducted using identical hyperparameters: a global batch size of 128, a learning rate of $2 \times 10^{-5}$, and the AdamW optimizer.

## F  EVALUATION DETAILS

### F.1  GENERATION METHODS

We implemented an efficient autoregressive inference pipeline for predicting keyboard and mouse actions from desktop screen captures or YouTube videos. Starting from MCAP files containing synchronized, timestamped data streams (screen captures and mouse/keyboard events), we resample the events at fixed intervals (50 ms for screen and mouse events, pass-through for keyboard inputs) and tokenize them as described in Appendix C. A dynamic context manager maintains a sliding window of recent events with efficient embedding caching, using a token context length of 2048. To accelerate inference, we apply several optimization techniques, including PyTorch model compilation, FlashAttention, and mixed-precision computation with bfloat16. For multi-GPU inference,

we leverage NVIDIA MPS. The generated token sequences are decoded back into structured MCAP events and subsequently evaluated. For pseudo-labeling YouTube videos, we generate MCAP files consisting of two-minute segments of screen events, excluding the first minute and last two minutes to mitigate the influence of introductions and outros.

Throughout this work, we evaluate the Generalist-IDM using fully autoregressive action decoding, both for the experiments in Section 5 and for pseudo-labeling YouTube videos. Teacher forcing was not used.

### F.2    EVALUATION METRICS

We evaluate the Generalist-IDM using fine-grained metrics that capture the correctness of predicted actions across both continuous (mouse) and discrete (keyboard, mouse button) modalities. All metrics operate on **non-overlapping 50 ms temporal bins** (corresponding to 20 Hz): within each bin, raw events are aggregated before comparison. Given an evaluation episode spanning $[t_{\text{start}}, t_{\text{end}})$, we partition the timeline into $n = \lceil (t_{\text{end}} - t_{\text{start}})/50\text{ms} \rceil$ bins. For each bin $i$, all mouse movement deltas $(\Delta x, \Delta y)$ are summed to obtain a single ground-truth vector $\mathbf{s}_i = (s_{i,x}, s_{i,y})$ and a predicted vector $\mathbf{d}_i = (d_{i,x}, d_{i,y})$.

**Pearson Correlation (X/Y).**    Pearson correlation measures the linear correspondence between ground-truth and predicted mouse displacement sequences, computed independently for the X and Y axes:

$$r_x = \frac{\sum_{i=1}^{n}(s_{i,x} - \bar{s}_x)(d_{i,x} - \bar{d}_x)}{\sqrt{\sum_{i=1}^{n}(s_{i,x} - \bar{s}_x)^2}\,\sqrt{\sum_{i=1}^{n}(d_{i,x} - \bar{d}_x)^2}}, \tag{6}$$

where $\bar{s}_x = \frac{1}{n}\sum_i s_{i,x}$ and $\bar{d}_x = \frac{1}{n}\sum_i d_{i,x}$ (analogously for $r_y$). A value of $1.0$ indicates perfect directional agreement; values near $0$ indicate no linear relationship. This metric captures whether the model predicts movement in the correct direction and with the correct relative timing, independent of magnitude.

**Scale Ratio (X/Y).**    Scale ratio quantifies the magnitude mismatch between ground-truth and predicted movements along each axis:

$$\text{ScaleRatio}_x = \frac{\frac{1}{n}\sum_{i=1}^{n}|s_{i,x}|}{\frac{1}{n}\sum_{i=1}^{n}|d_{i,x}|}. \tag{7}$$

To ensure interpretability, ratios $< 1$ are inverted so that all reported values satisfy ScaleRatio $\geq 1$. A value of $1.0$ indicates perfect magnitude match; larger values indicate that predictions are systematically scaled up or down relative to the ground truth. This metric is complementary to Pearson correlation: a model may achieve high correlation (correct direction) but poor scale ratio (wrong sensitivity), which is particularly relevant for mouse sensitivity calibration across games.

**Keyboard Accuracy.**    Keyboard accuracy measures whether the model correctly predicts the set of key events within each 50 ms bin. For each bin, we extract the multiset of key events, where each event is identified by its type (press/release) and virtual key code (e.g., `press_W`, `release_Shift`). Let $K_i^{\text{src}}$ and $K_i^{\text{pred}}$ denote the ground-truth and predicted multisets for bin $i$. For each unique key $k \in K_i^{\text{src}} \cup K_i^{\text{pred}}$, we assign a binary accuracy:

$$\text{acc}(k, i) = \begin{cases} 1 & \text{if } \text{count}(k, K_i^{\text{src}}) = \text{count}(k, K_i^{\text{pred}}), \\ 0 & \text{otherwise.} \end{cases} \tag{8}$$

The final keyboard accuracy is the mean over all key-bin pairs: $\text{KbdAcc} = \frac{1}{|\mathcal{K}|}\sum_{(k,i)\in\mathcal{K}} \text{acc}(k, i)$, where $\mathcal{K}$ is the set of all (key, bin) pairs with at least one event in either source or prediction.

**Mouse Button/Scroll Accuracy.**    Mouse button accuracy follows the same per-bin binary matching as keyboard accuracy, applied to button events (left/right/middle, down/up). Mouse scroll accuracy compares the total scroll amount within each bin, assigning binary correctness based on exact match.

# G  ADDITIONAL EVALUATION RESULTS

## G.1  ABLATION STUDY ON TEMPORAL OFFSETS

We further conduct a comprehensive ablation study to analyze the role of the temporal offset parameter $\tau$ in Generalist-IDM. We trained models with different temporal offsets, specifically $\tau \in \{0, 50, 100, 150, 200\}$ ms, and evaluated them on six in-distribution video games.

As shown in Tables 17 and 18, removing the temporal offset entirely ($\tau = 0$) leads to dramatic degradation across all metrics: Pearson correlations collapse to near zero, action-scale errors grow by more than an order of magnitude, and keypress accuracy drops sharply. This confirms that temporal misalignment severely harms multimodal action prediction and that NEP without future context is fundamentally insufficient.

Introducing a small offset ($\tau = 50$ ms) improves stability, but performance remains suboptimal, particularly for keypress prediction. This suggests that 50 ms does not provide enough future context for reliable behavior inference. Performance stabilizes once $\tau \geq 100$ ms, with all metrics converging to a high-performing regime and only minor variation between $\tau = 100$ ms and $\tau = 200$ ms. Notably, no single $\tau$ within this range consistently dominates, indicating that NEP-$\tau$ is robust to the exact offset choice as long as sufficient future context is provided. Based on these results, we adopt $\tau = 100$ ms as the default configuration in all experiments, balancing strong performance with practical responsiveness.

## G.2  TRAINING LOSS CURVES

To validate that desktop pretraining provides better initialization for embodied AI tasks, we analyze the training loss curves when fine-tuning on robot manipulation (LIBERO; Figure 7) and navigation (CANVAS; Figure 8) benchmarks, comparing the baseline (InternVL3-1B without desktop pretraining) against VAPT w/o pseudo. All curves are smoothed using an exponential moving average (EMA) with $\alpha = 0.10$ for clarity.

Across both manipulation and navigation settings, VAPT initialization leads to markedly improved optimization behavior:

- **Stable early-stage learning**: In LIBERO-Spatial and other benchmarks, the baseline exhibits a plateau at high loss values for approximately 1,000 steps, indicating the model must learn fundamental representations from scratch. In contrast, VAPT models show smooth, consistent loss reduction from the start.

- **Consistently lower loss**: Throughout training, VAPT maintains lower loss values compared to the baseline, suggesting better-aligned representations for embodied control tasks.

# H  DOWNSTREAM DETAILS

## H.1  ROBOT MANIPULATION

For LIBERO (Liu et al., 2023) evaluation, we train a manipulation policy identical to *openvla-oft* (Kim et al., 2025), except that the vision–language backbone is replaced with InternVL3-1B (or its OWA variant). The policy retains the L1 regression head for continuous action prediction, employs bidirectional attention in the policy stack, and uses parallel decoding with action chunking (chunk size $K = 8$).

The inputs consist of a third-person image, a wrist-camera image, the robot proprioceptive state, and a language instruction, resulting in two images per step (exocentric and egocentric). Training uses a filtered dataset where unsuccessful demonstrations are removed.

Optimization follows the *openvla-oft* recipe: LoRA rank 32, learning rate $5 \times 10^{-4}$, batch size 8, and image augmentation enabled. Linear decay is applied after 10,000 steps, with a total training budget of 15,005 steps. Checkpoints are saved every 1,000 steps, keeping both periodic and latest versions.

| Game | Model (ms) | Pearson | | Scale Ratio | | Keypress Acc. | |
|---|---|---|---|---|---|---|---|
| | | X | Y | X | Y | Kbd | Mouse |
| Apex | 0 | 0.301 | 0.168 | 1.71 | 13.48 | 0.584 | 0.997 |
| | 50 | 0.879 | 0.819 | 1.10 | 1.29 | 0.526 | 0.997 |
| | 100 | 0.839 | 0.853 | 1.13 | 1.23 | 0.765 | 0.997 |
| | 150 | 0.910 | 0.865 | 1.08 | 1.30 | 0.769 | 0.997 |
| | 200 | 0.897 | 0.793 | 1.28 | 2.24 | 0.760 | 0.998 |
| GTA | 0 | 0.101 | -0.098 | 1.28 | 24.66 | 0.636 | 0.972 |
| | 50 | 0.780 | 0.487 | 1.08 | 2.07 | 0.575 | 0.972 |
| | 100 | 0.794 | 0.839 | 1.09 | 1.42 | 0.698 | 0.941 |
| | 150 | 0.906 | 0.526 | 1.07 | 1.19 | 0.745 | 0.971 |
| | 200 | 0.920 | 0.782 | 1.27 | 1.42 | 0.734 | 0.879 |
| Minecraft | 0 | -0.016 | -0.007 | 13.06 | 50.40 | 0.077 | 0.925 |
| | 50 | 0.655 | 0.607 | 1.43 | 1.99 | 0.353 | 0.911 |
| | 100 | 0.803 | 0.784 | 1.24 | 1.27 | 0.610 | 0.917 |
| | 150 | 0.854 | 0.906 | 1.07 | 1.07 | 0.741 | 0.938 |
| | 200 | 0.908 | 0.895 | 1.19 | 1.16 | 0.753 | 0.933 |
| Brotato | 0 | 0.099 | 0.165 | 1.50 | 26.09 | 0.439 | 0.977 |
| | 50 | 0.962 | 0.938 | 1.03 | 1.03 | 0.456 | 0.980 |
| | 100 | 0.737 | 0.820 | 1.37 | 1.29 | 0.864 | 0.985 |
| | 150 | 0.963 | 0.953 | 1.07 | 1.04 | 0.873 | 0.990 |
| | 200 | 0.944 | 0.910 | 1.18 | 1.10 | 0.795 | 0.985 |
| Stardew | 0 | 0.139 | 0.102 | 4.18 | 14.67 | 0.229 | 0.960 |
| | 50 | 0.748 | 0.801 | 1.08 | 1.10 | 0.526 | 0.894 |
| | 100 | 0.830 | 0.756 | 1.13 | 1.17 | 0.744 | 0.964 |
| | 150 | 0.823 | 0.851 | 1.04 | 1.05 | 0.781 | 0.950 |
| | 200 | 0.796 | 0.768 | 1.39 | 1.58 | 0.704 | 0.942 |
| Core Keeper | 0 | 0.024 | -0.027 | 132.21 | 4.51 | 0.354 | 0.933 |
| | 50 | 0.793 | 0.755 | 1.29 | 1.38 | 0.514 | 0.935 |
| | 100 | 0.773 | 0.645 | 1.43 | 1.51 | 0.700 | 0.940 |
| | 150 | 0.888 | 0.805 | 1.17 | 1.19 | 0.690 | 0.943 |
| | 200 | 0.894 | 0.872 | 1.35 | 1.41 | 0.696 | 0.948 |

Table 17: Ablation on temporal offsets $\tau$ for in-domain games (0–200ms)

| $\tau$ (ms) | Avg Pearson X | Avg Pearson Y | Avg Scale X | Avg Scale Y | Avg Keypress (Kbd) | Avg Keypress (Mouse) |
|---|---|---|---|---|---|---|
| 0 | 0.108 | 0.050 | 25.66 | 22.30 | 0.386 | 0.961 |
| 50 | 0.803 | 0.734 | 1.17 | 1.48 | 0.492 | 0.948 |
| 100 | 0.796 | 0.783 | 1.23 | 1.31 | 0.730 | 0.957 |
| 150 | 0.891 | 0.818 | 1.08 | 1.14 | 0.767 | 0.965 |
| 200 | 0.893 | 0.837 | 1.28 | 1.49 | 0.740 | 0.947 |

Table 18: Aggregate performance across all games for different temporal offsets $\tau$. Per-game breakdown is provided in Appendix G.1.

Training is conducted on a single node with 8 GPUs via `torchrun`, with the same launch flags as *openvla-oft*, except for swapping the backbone to InternVL3-1B/OWA.

Evaluation is performed on the LIBERO benchmark (Liu et al., 2023), which includes four suites of manipulation tasks: (1) Spatial, varying scene layouts with fixed objects; (2) Object, varying the set of objects in a fixed scene; (3) Goal, testing goal-conditioned behavior; and (4) Long (LIBERO-10), long-horizon compositional tasks involving diverse objects, layouts, and goals. We report the average success rate over 500 episodes for each suite.

In the Meta-World (Yu et al., 2020) evaluation, we use the official LeRobot (Cadene et al., 2024) v0.4.1 codebase to train and evaluate the models across various tasks of different difficulty levels:

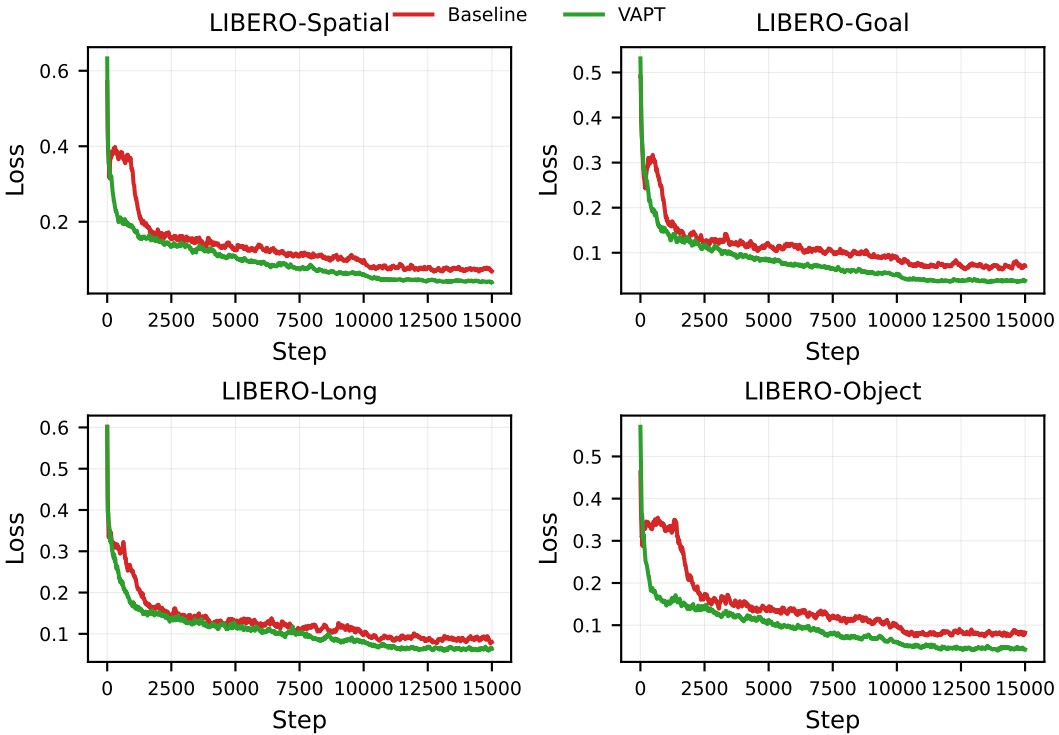

Figure 7: Training loss curves for all four LIBERO suites (Spatial, Goal, Long, Object). VAPT models consistently show immediate convergence without the initial plateau observed in the baseline.

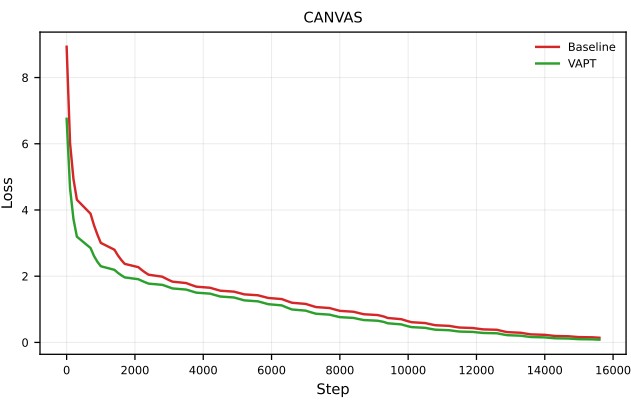

Figure 8: Training loss curves for CANVAS navigation tasks.

Easy, Medium, Hard, and Very Hard. The training process involves 50,000 steps with a learning rate of $1 \times 10^{-4}$, with all other hyperparameters left at default settings. The InternVL3-1B backbone is adapted for use with the VAPT model following the same protocol as SmolVLA (Shukor et al., 2025), without modifying any architecture parameters. Each task is evaluated using 10 episodes, and the success rates are reported for each difficulty level.

For the real-world evaluation, we follow the protocol used in SmolVLA (Shukor et al., 2025) for a pick-and-place task with the SO101 robot arm. The task is defined by the instruction: "Pick the blue cube and place it in the white box," with the cube placed at five distinct initial positions. We use the LeRobot (Cadene et al., 2024) framework for data collection, training, and evaluation. The setup includes two RGB cameras (top and side views), a green-screen background, and a fixed initial pose. A total of 208 demonstration episodes are collected, and both baseline and VAPT models are trained using the same downstream learning protocol. Each trained model is evaluated using 30

rollouts (two trials per cube position) with success measured as correctly grasping and placing the cube inside the box.

## H.2 ROBOT NAVIGATION

We established a baseline following CANVAS (Choi et al., 2024) by training an InternVL3-based model architecture on the COMMAND dataset. The baseline model was initialized with the default InternVL3 weights, whereas the VAPT w/o pseudo and VAPT w pseudo were trained from pretrained weights. All models were trained with full parameter unfreezing.

For optimization, we employed AdamW with separate learning rates: $2 \times 10^{-5}$ for the LLM, and $5 \times 10^{-5}$ for both the projector and vision encoder. Training was conducted with a batch size of 32 over 5 epochs, and each model utilized 128 waypoint tokens. In the main experiments, inference of CANVAS models was performed on a single NVIDIA H100 GPU. All evaluations were repeated three times per test dataset with randomized initial orientations.

## I ETHICS STATEMENT

We acknowledge and adhere to the ICLR Code of Ethics.

**Human Data Collection.** Our dataset was collected from 14 volunteer annotators who provided informed consent for gameplay recordings. Participants were fully informed about screen capture and input logging procedures and could withdraw at any time. All data underwent automated and manual review to remove any personally identifiable information before research use.

**Public Data Usage.** We processed only publicly available YouTube videos with permissive licenses for pseudo-labeling. Our focus on gaming content inherently minimizes privacy concerns compared to general desktop recording, as gaming interfaces rarely contain sensitive personal information.

**Transparency and Responsible Release.** To ensure responsible use, we will publicly release all code, data collection tools, and model weights with comprehensive documentation. We acknowledge that vision-action models could have dual-use potential; however, our focus on standardized gaming environments and transparent methodology helps mitigate misuse risks. Our computational approach (requiring only modest GPU resources) democratizes access while reducing environmental impact compared to large-scale training paradigms.

## J LIMITATIONS

While we validate our approach on both simulation benchmarks (LIBERO, Meta-World, CANVAS) and a real-world pick-and-place task with the SO-101 robot arm, the real-world evaluation remains limited to a single task setting. Broader real-robot validation across diverse manipulation and navigation scenarios is an important direction for future work. The differential impact of pseudo-labels (improving navigation but degrading manipulation) suggests task-specific transfer mechanisms that require further investigation. Our dataset focuses primarily on gaming interactions, which may not capture the full spectrum of desktop activities relevant to general-purpose robotics. Despite these constraints, our framework democratizes embodied AI research by reducing storage requirements by $152\times$ and training costs to under $1000, making large-scale vision-action pretraining accessible to resource-limited academic labs.

