# OpenReview forum: "D2E: Scaling Vision-Action Pretraining on Desktop Data for Transfer to Embodied AI"
_ICLR.cc/2026/Conference — ICLR 2026 Poster_

### Official Review · Reviewer_rEUi · 2025-10-24

**Soundness:** 2
**Presentation:** 2
**Contribution:** 3
**Rating:** 4
**Confidence:** 3

**Summary:**

This paper proposes D2E (Desktop-to-Embodied AI), a framework that leverages large-scale desktop interaction data (screen, keyboard, mouse) as an alternative to expensive real-world embodied trajectories for pretraining vision-action models.
The system consists of three main components: (1) OWA Toolkit – a high-performance desktop data collection and compression pipeline based on an extended MCAP format (OWAMcap), achieving up to 152× compression over prior datasets. (2)Generalist-IDM – a timestamp-aware inverse dynamics model (NEP-τ) that predicts human actions from videos and is used to pseudo-label over 1K hours of YouTube gameplay. (3)VAPT – a vision-action pretrained model that transfers desktop-learned representations to robot manipulation (LIBERO) and navigation (CANVAS), achieving 96.6% and 83.3% success rates respectively.

**Strengths:**

1. **Solid engineering contribution**: The OWA Toolkit is an impressive system-level effort that enables synchronized, multimodal desktop data collection and efficient storage. The compression performance and open-source reproducibility are highly commendable.

2. **Reproducibility and openness**: The authors provide thorough implementation details, training settings, and datasets, which greatly enhance the paper’s credibility and community value.

**Weaknesses:**

1. **Lack of academic novelty**:
The overall contribution is primarily engineering-oriented rather than conceptual or algorithmic. While the proposed OWA Toolkit and data infrastructure are impressive from a systems perspective, the work introduces limited new ideas in terms of representation learning, model design, or theoretical insight.

2. **Generalist-IDM design appears incremental**:
The core methodological component—Generalist Inverse Dynamics Model (Generalist-IDM)—is not sufficiently novel or deeply motivated. The NEP-τ formulation is essentially an incremental extension of standard next-event prediction, and its role in improving downstream performance remains unclear. It is uncertain whether the strong results of VAPT stem from the NEP-τ design itself or simply from the high-quality and diverse data collected by OWA Toolkit.

3. **Lack of ablation and sensitivity analysis**:
The paper provides limited empirical evidence dissecting the effectiveness of individual components. In particular, the temporal offset parameter τ is introduced as a key idea, yet there is no systematic study on how τ is selected or how sensitive the model performance is to its value. Without such analysis, the robustness and generality of the proposed modeling choice remain uncertain.

4. **Missing causal connection between design and outcome**:
Although the experiments show that a model trained on limited human desktop data and pseudo-labeled gameplay videos can generalize across unseen domains, the paper does not clearly explain why this happens. It lacks causal analysis connecting the proposed components (e.g., NEP-τ, OWA data quality, pseudo-labeling) to the observed generalization behavior. As a result, it is difficult to attribute the performance gains to specific methodological factors rather than data scale or diversity alone.

**Questions:**

1. Could the authors provide a deeper analysis of how the temporal offset mechanism (NEP-τ) contributes to the observed improvements? In particular, how do results change if τ is removed or varied?

2. To what extent do the downstream results of VAPT originate from the model design (NEP-τ, Generalist-IDM) versus the quality and diversity of the collected OWA data? Some controlled comparisons would help isolate these factors.

3. The paper shows strong zero-shot transfer to unseen games and robotic tasks. What is the hypothesized mechanism behind this generalization? Are certain components (e.g., timestamp-based tokenization) more critical than others?

---

> ### Author Response · Authors · 2025-11-25
>
> > Lack of academic novelty: The overall contribution is primarily engineering-oriented rather than conceptual or algorithmic. While the proposed OWA Toolkit and data infrastructure are impressive from a systems perspective, the work introduces limited new ideas in terms of representation learning, model design, or theoretical insight.
>
> We thank the reviewer for recognizing the system-level impressiveness of our OWA Toolkit and data infrastructure. We would like to offer a broader view on how our contributions—spanning data infrastructure, system design, and open reproducibility—align with the evolving priorities of the learning community.
>
> **1. Data and Systems as Core Enablers of Learning Research**
>
> While we acknowledge that our work does not introduce a new neural architecture, we believe that in the current era of foundation models, **scalable data pipelines and rigorous benchmarks are not merely engineering support but are central scientific contributions.**
> The ICLR community has increasingly recognized that overcoming systemic bottlenecks—such as data scarcity and evaluation methodology—is a prerequisite for algorithmic progress. This is evidenced by several recent *Oral* and *Spotlight* papers that focused primarily on data scaling, benchmarks, and system infrastructure:
>
> * **[1] Data Scaling Laws in Imitation Learning (ICLR 2025, Oral):** Established scaling laws for robotic manipulation, highlighting systematic data analysis as a key contribution.
> * **[2] RM-Bench (ICLR 2025, Oral) & [3] BigCodeBench (ICLR 2025, Oral):** Focused on rigorous evaluation frameworks for reward models and code generation, respectively.
> * **[4] MMAU (ICLR 2025, Spotlight) & [5] MathVista (ICLR 2024, Oral):** Contributed massive multi-task benchmarks that have catalyzed research in audio and mathematical reasoning.
>
> In this context, our **D2E framework** contributes the first scalable recipe for leveraging uncurated desktop data for embodied agents. We believe determining *how* to collect, process, and align this data effectively is a learning problem in itself, one that our work addresses empirically.
>
> **2. Commitment to Open Science and Community Impact**
>
> Moreover, we emphasize the value of our open-source contribution. Similar to **OLMoE (ICLR 2025 Oral) [6]**, which was recognized for its fully open, reproducible Mixture-of-Experts modeling, we believe that transparency in data and engineering is a critical academic contribution.
> * **Engineering for the Community:** Our **OWAMcap** format and the **OWA Toolkit** offer open-source solutions to several major bottlenecks in this field, which is detailed in the appendix: efficient data storage format with high compression rates, flexible media storage with supporting both embedded and external media files, highly performant screen capture program that integrates multiple tech stacks, and an efficient dataset format enough to support the I/O throughput required for training on massive video datasets. By open-sourcing these tools, we aim to lower the barrier to entry for other researchers, facilitating reproducible research in large-scale embodied AI.
>
> We hope the reviewer considers these contributions—validating a scalable data paradigm, providing efficient design choices, and enabling community research through open infrastructure—as a substantial and sufficient basis for acceptance at ICLR.
>
> ---
>
> References:
>
> [1] Lin et al., "Data Scaling Laws in Imitation Learning for Robotic Manipulation," ICLR 2025.
>
> [2] Liu et al., "RM-Bench: Benchmarking Reward Models of Language Models with Subtlety and Style," ICLR 2025.
>
> [3] Zhuo et al., "BigCodeBench: Benchmarking Code Generation with Diverse Function Calls and Complex Instructions," ICLR 2025.
>
> [4] Sakshi et al., "MMAU: A Massive Multi-Task Audio Understanding and Reasoning Benchmark," ICLR 2025.
>
> [5] Lu et al., "MathVista: Evaluating Mathematical Reasoning of Foundation Models in Visual Contexts," ICLR 2024.
>
> [6] Muennighoff et al., "OLMoE: Open Mixture-of-Experts Language Models," ICLR 2025.

---

> ### Author Response · Authors · 2025-11-25
>
> > Generalist-IDM design appears incremental: The core methodological component—Generalist Inverse Dynamics Model (Generalist-IDM)—is not sufficiently novel or deeply motivated. The NEP-τ formulation is essentially an incremental extension of standard next-event prediction, and its role in improving downstream performance remains unclear. (subset of weakness 2)
>
> We appreciate the reviewer’s perspective and understand the concern. We would like to clarify that the primary contribution of our work does not rely on the architectural novelty of the Generalist-IDM or NEP-τ. Instead, these components serve as *natural choices* that enable the broader contributions of the paper—showing how a simple, unified formulation paired with appropriate data alignment can unlock strong cross-domain generalization and emergent capabilities at scale.
>
> **1. Algorithmic Design: Intentional Simplicity and Efficiency**
>
> Our goal was not to propose a complex new architecture, but to identify the **most effective and efficient design choice** for cross-domain transfer.
>
> * **NEP as a Natural Choice:** We intentionally avoid architectural modifications (e.g., separate action encoders or decoders) to maintain flexibility and to fully exploit the pretrained knowledge embedded in MLLMs. Event-based modeling via NEP was chosen to efficiently capture long-horizon dependencies (up to 8k context length), while allowing standard MLLMs to operate purely via tokenization.
>
> * **NEP-τ as a Natural Extension:** Since the original NEP is simply the τ = 0 case of NEP-τ, introducing a temporal offset τ to incorporate immediate future observations becomes a natural and minimal generalization. Unlike traditional IDMs that often require specialized action heads or fixed-interval assumptions, our approach aligns the asynchronous nature of human inputs while preserving the natural token-based structure of standard MLLMs. By treating time and action as tokens, we can utilize off-the-shelf MLLM backbones *as is,* preserving their autoregressive attention mechanisms without the need for complex auxiliary encoders.
>
> **2. Novelty in Scope and Emergent Behaviors**
>
> The primary novelty of the Generalist-IDM lies in its **application scope** rather than architectural complexity. To the best of our knowledge, prior IDM works train models within a single environment or, at most, a small set of closely related ones [7][8][9]. In contrast, we train a single Generalist-IDM across a **highly diverse collection of desktop tasks.**
> This expanded scope leads to important empirical findings: a simple inverse-dynamics objective, when scaled with sufficiently diverse data, gives rise to **emergent Out-of-Distribution (OOD) generalization and in-context adaptation.** Identifying and validating that such behaviors arise *without modifying the architecture* represents a meaningful advance for the field.
>
> ---
>
> References:
>
> [7] Baker et al., "Video pretraining (vpt): Learning to act by watching unlabeled online videos,” NeurIPS 2022
>
> [8] Du et al., “Learning Universal Policies via Text-Guided Video Generation,” NeurIPS 2023
>
> [9] Bjorck et al., “GR00T N1: An Open Foundation Model for Generalist Humanoid Robots,” Arxiv 2025

---

> ### Author Response · Authors · 2025-11-25
>
> > The NEP-τ formulation is essentially an incremental extension of standard next-event prediction, and its role in improving downstream performance remains unclear. (subset of weakness 2)
>
> > Lack of ablation and sensitivity analysis: The paper provides limited empirical evidence dissecting the effectiveness of individual components. In particular, the temporal offset parameter τ is introduced as a key idea, yet there is no systematic study on how τ is selected or how sensitive the model performance is to its value. Without such analysis, the robustness and generality of the proposed modeling choice remain uncertain.
>
> > Could the authors provide a deeper analysis of how the temporal offset mechanism (NEP-τ) contributes to the observed improvements? In particular, how do results change if τ is removed or varied?
>
> We sincerely appreciate the reviewer’s careful attention to the temporal offset mechanism, and we agree that a deeper analysis is valuable. To systematically investigate the role of the temporal offset parameter τ, we trained the Generalist-IDM with τ ∈ {0, 50, 100, 150, 200} ms and evaluated the resulting models on six in-distribution video games. Note that τ = 0 ms corresponds to removing the temporal offset entirely (i.e., standard NEP without future observation).
>
> **Table: Performance on 2D and 3D Games Across Different Temporal Offsets (τ)**
>
> |Game|τ(ms)|Pearson X|Pearson Y|Scale X|Scale Y|Keypress(Kbd)|Keypress(Mouse)|
> |-|-|-|-|-|-|-|-|
> |**Apex**|0|0.301|0.168|1.71|13.48|0.584|0.997|
> ||50|0.879|0.819|1.10|1.29|0.526|0.997|
> ||100|0.839|0.853|1.13|1.23|0.765|0.997|
> ||150|0.910|0.865|1.08|1.30|0.769|0.997|
> ||200|0.897|0.793|1.28|2.24|0.760|0.998|
> |**GTA**|0|0.101|-0.098|1.28|24.66|0.636|0.972|
> ||50|0.780|0.487|1.08|2.07|0.575|0.972|
> ||100|0.794|0.839|1.09|1.42|0.698|0.941|
> ||150|0.906|0.526|1.07|1.19|0.745|0.971|
> ||200|0.920|0.782|1.27|1.42|0.734|0.879|
> |**Minecraft**|0|-0.016|-0.007|13.06|50.40|0.077|0.925|
> ||50|0.655|0.607|1.43|1.99|0.353|0.911|
> ||100|0.803|0.784|1.24|1.27|0.610|0.917|
> ||150|0.854|0.906|1.07|1.07|0.741|0.938|
> ||200|0.908|0.895|1.19|1.16|0.753|0.933|
> |**Brotato**|0|0.099|0.165|1.50|26.09|0.439|0.977|
> ||50|0.962|0.938|1.03|1.03|0.456|0.980|
> ||100|0.737|0.820|1.37|1.29|0.864|0.985|
> ||150|0.963|0.953|1.07|1.04|0.873|0.990|
> ||200|0.944|0.910|1.18|1.10|0.795|0.985|
> |**Stardew**|0|0.139|0.102|4.18|14.67|0.229|0.960|
> ||50|0.748|0.801|1.08|1.10|0.526|0.894|
> ||100|0.830|0.756|1.13|1.17|0.744|0.964|
> ||150|0.823|0.851|1.04|1.05|0.781|0.950|
> ||200|0.796|0.768|1.39|1.58|0.704|0.942|
> |**Core Keeper**|0|0.024|-0.027|132.21|4.51|0.354|0.933|
> ||50|0.793|0.755|1.29|1.38|0.514|0.935|
> ||100|0.773|0.645|1.43|1.51|0.700|0.940|
> ||150|0.888|0.805|1.17|1.19|0.690|0.943|
> ||200|0.894|0.872|1.35|1.41|0.696|0.948|
>
> **Table: Aggregate Metrics Across All Games for Different Temporal Offsets (τ)**
>
> |τ(ms)|Avg Pearson X|Avg Pearson Y|Avg Scale X|Avg Scale Y|Avg Keypress(Kbd)|Avg Keypress(Mouse)|
> |-|-|-|-|-|-|-|
> |0|0.108|0.050|25.66|22.30|0.386|0.961|
> |50|0.803|0.734|1.17|1.48|0.492|0.948|
> |100|0.796|0.783|1.23|1.31|0.730|0.957|
> |150|0.891|0.818|1.08|1.14|0.767|0.965|
> |200|0.893|0.837|1.28|1.49|0.740|0.947|
>
> Analysis:
>
> **τ = 0 shows a critical performance drop.** Across all games, τ = 0 consistently shows severe degradation: Pearson correlations drop near zero (avg: 0.108 for X, 0.050 for Y), scale errors explode (avg: 25.66× for X, 22.30× for Y), and keypress accuracy plummets (avg: 0.386 vs. 0.73–0.77 for τ ≥ 100). This confirms that temporal alignment is essential.
>
> **τ = 50 shows partial recovery but remains suboptimal.** At τ = 50 ms, Pearson correlations improve significantly (0.803 for X, 0.734 for Y) and scale errors normalize (1.17×, 1.48×), but keypress accuracy remains notably lower (0.492 vs. 0.73–0.77 for τ ≥ 100). This suggests 50 ms provides insufficient future context for reliable action prediction.
>
> **τ ≥ 100 shows stable performance.** Once τ ≥ 100 ms, performance stabilizes with no strong trend across 100–200 ms. Aggregated results show Pearson X: 0.796 (100ms), 0.891 (150ms), 0.893 (200ms); Pearson Y: 0.783, 0.818, 0.837; Keypress (Kbd): 0.730, 0.767, 0.740. While individual metrics vary slightly, the differences are modest, and no single value consistently dominates across all metrics. The model is robust to τ selection within this range.
>
> These results indicate that NEP-τ requires a sufficient temporal offset (τ ≥ 100 ms) for stable performance. In our experiments, we used **τ = 100 ms** as the default configuration, which provides strong and stable performance across all metrics. The ablation confirms that the exact choice within the 100–200 ms range is not critical, demonstrating the robustness of our approach.

---

> ### Author Response · Authors · 2025-11-25
>
> > It is uncertain whether the strong results of VAPT stem from the NEP-τ design itself or simply from the high-quality and diverse data collected by OWA Toolkit. (subset of weakness 2)
>
> > Missing causal connection between design and outcome: Although the experiments show that a model trained on limited human desktop data and pseudo-labeled gameplay videos can generalize across unseen domains, the paper does not clearly explain why this happens. It lacks causal analysis connecting the proposed components (e.g., NEP-τ, OWA data quality, pseudo-labeling) to the observed generalization behavior. As a result, it is difficult to attribute the performance gains to specific methodological factors rather than data scale or diversity alone.
>
> > To what extent do the downstream results of VAPT originate from the model design (NEP-τ, Generalist-IDM) versus the quality and diversity of the collected OWA data? Some controlled comparisons would help isolate these factors.
>
> > The paper shows strong zero-shot transfer to unseen games and robotic tasks. What is the hypothesized mechanism behind this generalization? Are certain components (e.g., timestamp-based tokenization) more critical than others?
>
> We appreciate the reviewer’s thoughtful question and fully agree on the importance of understanding which factors most strongly contribute to VAPT’s performance.
>
> Before addressing the question, we would like to clarify that **NEP-τ is used only when training the Generalist-IDM** to generate pseudo-labels from YouTube videos. In contrast, **VAPT itself does not use NEP-τ** and instead performs standard vision–action pretraining on the resulting human-labeled and pseudo-labeled trajectories.
>
> With this clarification, the reviewer’s question can be answered along two axes:
> * **1. Which components contribute to the strong performance?**
> * **2. Why does desktop-to-embodied transfer work?**
> ---
> **1. Which Components Contribute to Strong Performance?**
>
> VAPT’s performance arises from a combination of model-level and data-level factors:
>
> * **(a) Base Model and Architecture**: We use **InternVL3-1B**, a strong open-source VLM that already achieves competitive performance on its own (e.g., 84.8% on LIBERO and 75.3% on CANVAS). However, the substantial gains observed after training with VAPT (LIBERO: 84.8% → 96.6% (+11.8pp) and CANVAS: 75.3% → 83.3% (+8.0pp)) indicate that the improvements do not come from the base model alone but instead arise from the VAPT training process.
>
> * **(b) Data: Quality, Quantity, and Diversity**: Our experiments provide evidence of the roles of these factors.
> Section 5.1 and Tables 3–5 show that **Generalist-IDM (20 games)** consistently outperforms Specialist-IDMs, indicating that **data diversity** is important.
> Tables 6–7 show task-dependent patterns: manipulation (LIBERO) is more sensitive to **label quality** (VAPT w/o pseudo: 96.6% → w/ pseudo: 92.2%, +4.4pp), while navigation (CANVAS) benefits more from **data quantity and diversity** (VAPT w/o pseudo: 75.3% → VAPT w/ pseudo 83.3%, +8.0pp).
>
> **2. Why Does Desktop-to-Embodied Transfer Work?**
>
> Although we do not provide a complete causal decomposition, our findings consistently reveal three concrete factors that distinguish VAPT from the baseline:
>
> * **(1) Action Modality Alignment**: The baseline VLM is pretrained only on image–text or video–text data, lacking explicit action supervision. In contrast, VAPT provides a pretraining procedure with explicit action modality, where human demonstrations and pseudo-labeled gameplay supply vision–action trajectories that align naturally with robotic control tasks.
>
> * **(2) Skills for Goal-Directed Sequential Decision-Making**: Desktop gameplay requires skills that transfer to robotics: visual grounding (identifying task-relevant objects and spatial relationships), temporal reasoning (planning multi-step action sequences), goal-directed behavior (inferring task objectives and executing actions), and long-range dependencies (NEP training uses long context up to 8k tokens).
>
> * **(3) High Diversity of Desktop Game Data**: Video games exhibit diversity across visual (20 games spanning 2D/3D graphics, various art styles, and scene compositions), behavioral (different game mechanics, control schemes, and task structures), and rule (varied objectives, constraints, and success criteria) dimensions. While robotic tasks are not a strict subset of desktop gameplay, empirical results (LIBERO 96.6%, CANVAS 83.3%) suggest useful coverage.
>
> **Overall**
>
> We believe the τ ablation, the pseudo-label ablation, and the generalist vs. specialist comparisons already provide substantial causal insight into where VAPT’s performance comes from. Our primary contribution is demonstrating desktop-to-embodied transfer across diverse benchmarks (LIBERO, CANVAS, Meta-World) and ongoing real-world experiments. We are happy to provide additional analysis if the reviewer has specific leftover concerns.

---

### Official Review · Reviewer_SNK1 · 2025-10-29

**Soundness:** 2
**Presentation:** 2
**Contribution:** 2
**Rating:** 4
**Confidence:** 3

**Summary:**

The paper introduces D2E, a framework that leverages large-scale desktop interactions collected via the OWA toolkit and generalized through a timestamp-based IDM for pretraining embodied AI. Using more than 1.3K hours of human and pseudo-labeled gameplay data, the approach demonstrates strong transfer to robotics tasks, achieving 96.6% on LIBERO manipulation and 83.3% on CANVAS navigation.

**Strengths:**

1. Using game data for embodied pretraining is an interesting direction.

2. The paper provides a detailed system design for data collection.

**Weaknesses:**

1. The main contribution of the paper lies in how to collect action data and in the system-level design, while the algorithmic innovation is relatively limited. Currently, there is a large body of work that uses OOD data for pretraining, so my concern is that this paper may not be a very good fit for a learning-focused conference.

2. Using game data for navigation tasks in CANVAS makes sense, but its effectiveness for manipulation tasks remains questionable. In addition, LIBERO can be easily hacked with certain tricks, which makes it difficult to serve as a fair or reliable benchmark for validating algorithms.

**Questions:**

Did the pretraining stage also include robotics data, or was it purely based on desktop game data?

---

> ### Author Response · Authors · 2025-11-24
>
> >The main contribution of the paper lies in how to collect action data and in the system-level design, while the algorithmic innovation is relatively limited. Currently, there is a large body of work that uses OOD data for pretraining, so my concern is that this paper may not be a very good fit for a learning-focused conference.
>
> **1. Data-Centric Contributions as Core Enablers of Learning Research**
>
> We respectfully appreciate the reviewer’s perspective but would like to clarify that data- and system-level contributions are not only welcome at ICLR, but have recently been *central* to the conference’s most impactful works. ICLR has increasingly emphasized that **rigorous data pipelines, scalable infrastructure, and benchmark design are essential enablers of learning progress**, forming the empirical foundation upon which algorithmic advances are built.
>
> This is reflected in multiple *Oral* and *Spotlight* papers at ICLR 2025 and 2024 whose primary contributions were data- or system-oriented:
>
> - [1] Data Scaling Laws in Imitation Learning for Robotic Manipulation (ICLR 2025, Oral): Introduces scaling-law analyses and a new robotic imitation benchmark—demonstrating that systematic data exploration is a highly valued research direction.
> - [2] RM-Bench (ICLR 2025, Oral): Presents a benchmark for evaluating reward models, highlighting that high-quality evaluation protocols are as critical as algorithmic novelty.
> - [3] BigCodeBench (ICLR 2025, Oral): Proposes a large-scale dataset and evaluation framework for code generation, recognized for its community impact rather than algorithmic complexity.
> - [4] MMAU (ICLR 2025, Spotlight): Introduces a massive multi-task audio understanding benchmark that catalyzes multimodal research.
> - [5] MathVista (ICLR 2024, Oral): Presents a large-scale benchmark for evaluating mathematical reasoning of foundation models, now exceeding 1,000 citations in under two years.
>
> These examples demonstrate that ICLR explicitly values contributions addressing **systemic bottlenecks**—such as data scalability, synchronization, and evaluation reproducibility—that *enable* new forms of learning. In the same spirit, our work introduces both (i) the first open and scalable **OWA Toolkit** for desktop-to-embodied data generation, and (ii) a validated **D2E (Desktop-to-Embodied) transfer methodology** that systematically quantifies how desktop-scale data can generalize to embodied settings. These contributions directly advance the data foundations of embodied AI, aligning closely with ICLR’s evolving scope.
>
> ---
>
> **2. Methodological Soundness and the Principle of “Simplicity by Design”**
>
> We further acknowledge the comment on “limited algorithmic innovation” and would like to clarify that **our methodological simplicity is intentional and scientifically motivated**, rather than a limitation.
>
> Our goal is to *isolate and quantify* the effect of large-scale, asynchronous desktop data on embodied learning. To achieve this, we deliberately employ a standard model backbone while introducing two key technical elements that make this possible:
>
> - **Event-Based Tokenization**, which discretizes human desktop interactions (mouse, keyboard, window events) into temporally aligned tokens with minimal information loss; and
> - **NEP-τ (Next-Event Prediction with Temporal Offset)**, a novel formulation that enables precise alignment between asynchronous human inputs and high-frequency visual observations.
>
> These innovations address the non-trivial challenge of representing high-rate, multi-modal human-computer interaction data in a form usable by standard transformer architectures—thereby extending existing LLM backbones to a new data regime without architectural overfitting.
>
> This design choice ensures that observed performance gains can be *attributed to data quality and scale*, not to confounding architectural changes. Such disciplined isolation of causal factors is essential for drawing scientifically meaningful conclusions about data scaling laws—precisely the kind of evidence-driven methodology that ICLR consistently rewards.
>
> ---
>
> References
>
> [1] Lin, Fanqi, et al. "Data Scaling Laws in Imitation Learning for Robotic Manipulation." The Thirteenth International Conference on Learning Representations.
>
> [2] Liu, Yantao, et al. "RM-Bench: Benchmarking Reward Models of Language Models with Subtlety and Style." The Thirteenth International Conference on Learning Representations.
>
> [3] Zhuo, Terry Yue, et al. "BigCodeBench: Benchmarking Code Generation with Diverse Function Calls and Complex Instructions." *The Thirteenth International Conference on Learning Representations*.
>
> [4] Sakshi, S., et al. "MMAU: A Massive Multi-Task Audio Understanding and Reasoning Benchmark." *The Thirteenth International Conference on Learning Representations*.
>
> [5] Lu, Pan, et al. "MathVista: Evaluating Mathematical Reasoning of Foundation Models in Visual Contexts." The Twelfth International Conference on Learning Representations.

---

> ### Author Response · Authors · 2025-11-24
>
> > Using game data for navigation tasks in CANVAS makes sense, but its effectiveness for manipulation tasks remains questionable. In addition, LIBERO can be easily hacked with certain tricks, which makes it difficult to serve as a fair or reliable benchmark for validating algorithms.
> >
>
> We thank the reviewer for highlighting the limitations of the LIBERO benchmark. We agree that while LIBERO is a useful starting point, it can be susceptible to shortcuts, and verifying our method on a more established manipulation benchmark is necessary to prove the transferability of desktop-learned representations.
>
> **1. Evaluation on Meta-World [6]**
>
> To address the concern regarding manipulation tasks, we evaluated our VAPT models on **Meta-World**, a standard benchmark for multi-task robotic manipulation. We compared VAPT against the InternVL3-1B baseline across tasks of varying difficulty.
>
> **Table: Success Rates on Meta-World Benchmark**
>
> | **Method** | **Easy** | **Medium** | **Hard** | **Very Hard** | **Avg** |
> | --- | --- | --- | --- | --- | --- |
> | Baseline (InternVL3-1B) | 55.4 | 14.5 | 1.7 | 8.0 | **19.9** |
> | VAPT (w/o pseudo) | 53.6 | 18.2 | 8.3 | 20.0 | **25.0** |
> | VAPT (w/ pseudo) | 52.1 | 16.4 | 6.7 | 24.0 | **24.8** |
>
> Implementation Details:
>
> To ensure a fair comparison, we used the official LeRobot v0.4.1 codebase. We adapted the InternVL3-1B backbone following the exact protocol used for SmolVLA, without modifying any architecture parameters. Both the baseline and VAPT models were trained for 50k steps with a learning rate of 1e-4, keeping all other hyperparameters at default settings. We conducted 10 evaluation episodes per task.
>
> Analysis:
>
> Even without robotics-specific pretraining or extensive hyperparameter tuning, VAPT consistently outperforms the baseline, showing an average success rate improvement of roughly 5% (a ~25% relative gain). This gap is most pronounced in the Hard and Very Hard categories (e.g., 8.0% vs. 20.0–24.0% on Very Hard), suggesting that the priors learned from desktop data are particularly robust for complex manipulation challenges.
>
> **2. Real-World Validation on SO-101 Arm**
>
> Beyond simulation, we are actively validating VAPT on a physical SO-101 robotic arm.
>
> - **Qualitative Results:** We have uploaded a demo video to the **anonymized website** linked in the abstract. This video demonstrates the model's capability to transfer desktop-learned priors to physical hardware.
> - **Quantitative Evaluation:** We are currently finalizing the quantitative metrics for these real-world tasks and will update the paper and the anonymous website with these results within the next few days.
>
> **3. Visualizing the Transfer Mechanism**
>
> Finally, to address the question of *why* desktop data is effective for manipulation despite the domain gap, we are currently working on visualizing the latent representations of our model. We aim to empirically show the alignment of sensorimotor primitives between desktop and robot trajectories in the feature space. We plan to update these visualization results to the paper within the next few days to provide further intuition on the transfer mechanism.
>
> We believe the combination of Meta-World results, real-world demonstrations, and the upcoming latent analyses addresses the concerns regarding benchmark reliability and the validity of desktop-to-robot transfer.
>
> Reference:
>
> [6] Yu et al., "Meta-World: A Benchmark and Evaluation for Multi-Task and Meta Reinforcement Learning," CoRL, 2019.

---

> ### Author Response · Authors · 2025-11-24
>
> > Did the pretraining stage also include robotics data, or was it purely based on desktop game data?
>
> Our VAPT **pretraining stage uses only desktop game data and does not include any robotics datasets**.
> As described in the paper, “pretraining” refers specifically to training on the large-scale vision–action dataset derived from desktop environments, where we generate pseudo-labels at scale.
>
> Robotics datasets such are used only during the finetuning stage, not during pretraining. This separation is intentional:
>
> - Our goal is to demonstrate that large-scale pseudo-labeled desktop data alone can provide strong general-purpose visuomotor representations.
> - Robotics-specific data is then used in finetuning to specialize the model for downstream manipulation or control tasks.
>
> This design also highlights the key benefit of **VAPT**: it enables scalable pretraining without requiring access to large real-world robotics datasets, which are expensive and difficult to collect.

---

> > ### Comment · Reviewer_SNK1 · 2025-11-27
> >
> > Thanks for your explanation. My questions have been answered. I will revise my score.

---

### Official Review · Reviewer_cyb3 · 2025-10-31

**Soundness:** 3
**Presentation:** 2
**Contribution:** 3
**Rating:** 6
**Confidence:** 2

**Summary:**

This paper presents D2E, a framework that scales vision-action pretraining using large-scale desktop interaction data (screen, keyboard, mouse) instead of costly physical robot trajectories. It introduces the OWA Toolkit for efficient multimodal data capture, an IDM for pseudo-labeling internet gameplay videos, and a VAPT that transfers learned representations to robotics. Trained on about 1.3K hours of human and pseudo-labeled data, the model achieves 96.6% success on libero and 83.3% on canvas navigation, showing that desktop interactions can effectively serve as scalable pretraining for embodied AI.

**Strengths:**

1. End-to-end pipeline and scale: The paper provides a complete framework covering data collection, pseudo-labeling, pretraining, and robotic transfer. It builds on 31 desktop games with 335 hours of human demonstrations and expands to over 1,000 hours of pseudo-labeled YouTube gameplay, achieving clear scalability.

2. Solid engineering contribution: The OWA toolkit enables synchronized screen, keyboard, and mouse capture with nanosecond precision and 152× compression, significantly reducing random-access I/O cost and improving data-loading throughput during training.

3. Openness and reproducibility: The authors commit to releasing the toolkit, datasets, and pretrained models with detailed documentation, ensuring transparency and easy reproducibility for future research.

**Weaknesses:**

The experimental evidence supporting why and when desktop data benefits embodied tasks remains insufficient. While results show positive transfer, the paper lacks a deeper analysis of task suitability—for example, whether desktop-derived sensorimotor patterns genuinely align with the fine-grained control and contact dynamics required in manipulation tasks.

**Questions:**

Please see the weaknesses.

---

> ### Author Response · Authors · 2025-11-25
>
> > The experimental evidence supporting why and when desktop data benefits embodied tasks remains insufficient. While results show positive transfer, the paper lacks a deeper analysis of task suitability—for example, whether desktop-derived sensorimotor patterns genuinely align with the fine-grained control and contact dynamics required in manipulation tasks.
>
> We thank the reviewer for raising this important question and fully agree on the need for deeper analysis of task suitability. Although we do not provide a complete causal decomposition, our findings consistently reveal three concrete factors that distinguish VAPT from the baseline:
>
> * **(1) Action Modality Alignment**: The baseline VLM is pretrained only on image–text or video–text data, lacking explicit action supervision. In contrast, VAPT provides a pretraining procedure with explicit action modality, where human demonstrations and pseudo-labeled gameplay supply vision–action trajectories that align naturally with robotic control tasks.
>
> * **(2) Skills for Goal-Directed Sequential Decision-Making**: Desktop gameplay requires skills that transfer to robotics: visual grounding (identifying task-relevant objects and spatial relationships), temporal reasoning (planning multi-step action sequences), goal-directed behavior (inferring task objectives and executing actions), and long-range dependencies (NEP training uses long context up to 8k tokens).
>
> * **(3) High Diversity of Desktop Game Data**: Video games exhibit diversity across visual (20 games spanning 2D/3D graphics, various art styles, and scene compositions), behavioral (different game mechanics, control schemes, and task structures), and rule (varied objectives, constraints, and success criteria) dimensions. While robotic tasks are not a strict subset of desktop gameplay, empirical results (LIBERO 96.6%, CANVAS 83.3%) suggest useful coverage.
>
> **Addressing the Reviewer's Specific Concern on Manipulation Suitability**
>
> Regarding whether desktop-derived sensorimotor patterns genuinely align with fine-grained control and contact dynamics required in manipulation tasks, our empirical results suggest that they do, although we acknowledge that a full mechanistic characterization remains future work.
>
> For training VAPT, our desktop data is resampled to 20Hz, matching the **20Hz control frequency** used in LIBERO’s manipulation tasks. This temporal alignment stabilizes the sensorimotor feedback loop by keeping the timing of observations, action sequences, and contact events consistent across domains. Under this setup, the baseline model achieves 84.8% on LIBERO, whereas **VAPT trained on desktop human demonstrations reaches 96.6% (+11.8pp).**
>
> This substantial improvement indicates that the representations learned from desktop gameplay provide transferable cues for fine-grained manipulation, even when the action spaces differ significantly (keyboard/mouse vs. 7-DoF robotic arm).
>
> **Scope of Our Contribution**
>
> Our main contribution is **demonstrating that desktop-to-embodied transfer works at scale** across multiple benchmarks (LIBERO, CANVAS, Meta-World) and providing the first open, scalable infrastructure (OWA Toolkit) to enable this research direction. While a deeper causal analysis of *why* this transfer occurs—such as identifying which specific sensorimotor primitives align across domains—is an important avenue for future work, our empirical findings establish a strong foundation for pursuing such analyses.
>
> We are currently conducting real-world manipulation experiments on a physical robotic arm and will share results during rebuttal.

---

> ### Comment · Reviewer_cyb3 · 2025-11-25
>
> Thanks for your rebuttal. I want to clarify that I do not question the value of this work within the Desktop Games area. This area is simply not my expertise.
>
> The real thing I am confused about is the motivation and the underlying logic behind this paradigm of transferring a desktop-pretrained model to embodied manipulation, which is one of the central contributions of the paper. If this pathway is truly effective, it could represent a new direction for improving manipulation. But at the moment, I still struggle to understand the logic behind this transfer.
>
> Therefore, I hope the authors can further clarify the following points rather than just demonstrating the improvement:
>
> **(1) What is the actual motivation for using desktop pre-training data for embodied manipulation tasks?**
>
> **(2) Why should such cross-domain transfer work in principle?**

---

> > ### Author Response · Authors · 2025-11-27
> >
> > We thank the reviewer for the clarification and appreciate the opportunity to address these fundamental questions about desktop-to-embodied transfer.
> >
> >
> > > (1) What is the actual motivation for using desktop pre-training data for embodied manipulation tasks?
> >
> > **The primary motivation is the dramatically lower cost and effort required for data collection compared to robot manipulation data.**
> >
> > Collecting large-scale robot manipulation datasets is extremely resource-intensive. For example:
> >
> > - **DROID** [1]: 350 hours of data required **50 data collectors** across **13 institutions** over **12 months**
> > - **Open X-Embodiment** [2]: Collaboration between **21 institutions** to assemble datasets from 22 different robots
> >
> > In contrast, our approach required:
> > - **Human demonstrations (335 hours across 31 games)**: Collected in **1 month** by **14 annotators** under **1 manager**
> > - **Pseudo-labeled data (1055 hours)**: Generated from **manually curated YouTube gameplay videos** using our Generalist-IDM. While not detailed in the paper, the curation process involved **1 researcher** reviewing and selecting videos with over **1 week**, requiring **no additional human annotation** for action labeling
> >
> > This represents a **substantial reduction** in both time and human effort compared to equivalent-scale robot data collection. Desktop data collection eliminates the need for:
> > - Physical robot hardware and maintenance
> > - Expert teleoperation or kinesthetic teaching skills
> > - Controlled lab environments with safety infrastructure
> > - Multi-institution coordination and standardization
> > - Robot-specific data collection protocols
> >
> > Furthermore, the ability to automatically pseudo-label internet-scale gameplay videos (1055 hours from YouTube) provides a scalable pathway to expand training data without proportional increases in human effort-a capability unavailable for robot manipulation data.
> >
> >
> > > (2) Why should such cross-domain transfer work in principle?
> >
> > **We identify three key factors that distinguish VAPT from the baseline VLM:**
> >
> > 1. **Action modality alignment**: The baseline is pretrained only on image-text data, while VAPT provides explicit vision-action trajectories
> > 2. **Goal-directed sequential decision-making skills**: Desktop gameplay requires visual grounding, temporal reasoning, and long-range dependencies that transfer to robotic control
> > 3. **High diversity**: 31 games spanning 2D/3D graphics, various mechanics, and task structures provide broad coverage
> >
> > **To provide supporting evidence for this hypothesis, we have added training loss curve analysis in the revised manuscript (Appendix E.1).** When fine-tuning on robot manipulation (LIBERO) and navigation (CANVAS) benchmarks, VAPT-initialized models exhibit substantially better training stability compared to the baseline:
> >
> > - **Stable early-stage learning**: In LIBERO-Spatial and other benchmarks, the baseline exhibits a plateau at high loss values for approximately 1,000 steps, indicating the model must learn fundamental representations from scratch. In contrast, VAPT models show smooth, consistent loss reduction from the start.
> > - **Consistently lower loss**: Throughout training, VAPT maintains lower loss values compared to the baseline, suggesting better-aligned representations for embodied control tasks.
> >
> > Training loss curves (Appendix E.1) provide supporting evidence: VAPT-initialized models show immediate convergence without the initial plateau observed in the baseline, suggesting better-aligned representations for embodied control. However, a complete mechanistic understanding remains an open question for future work.
> >
> > **References**
> >
> > [1] Khazatsky et al., "DROID: A Large-Scale In-The-Wild Robot Manipulation Dataset," RSS, 2024.
> >
> > [2] Open X-Embodiment Collaboration, "Open X-Embodiment: Robotic Learning Datasets and RT-X Models," CoRL, 2023.

---

### Official Review · Reviewer_5KYr · 2025-10-31

**Soundness:** 3
**Presentation:** 3
**Contribution:** 3
**Rating:** 8
**Confidence:** 4

**Summary:**

This paper proposes D2E, a framework for pretraining models on desktop data (e.g., game videos paired with human actions). D2E offers a comprehensive toolbox that includes standardized data collection and storage, efficient data reading, and a scalable pseudo-label annotation strategy for desktop data, powered by a specifically designed and trained Inverse Dynamics Model (IDM). Extensive experiments are conducted to validate the framework across multiple aspects, including data collection, processing, and reading efficiency, as well as the effectiveness of using D2E for pretraining embodied models. The results show that adapting the pretrained models to embodied and navigation tasks leads to notable performance improvements, revealing a promising direction for leveraging scalable desktop data in embodied model pretraining.

**Strengths:**

1. Given the high cost of embodied data collection, exploring new data sources that capture human intention knowledge and can contribute to embodied learning is a highly meaningful research direction. While desktop data has been explored in previous works for training embodied models, those methods primarily focus on developing and validating models within in-domain game environments. This paper extends beyond that scope, aiming to make desktop data more broadly useful for embodied learning.

2. The high-quality datasets collected and the well-designed data collection pipeline represent valuable contributions to the research community. They provide essential infrastructure for future work on large-scale embodied pretraining, facilitating both reproducibility and scalability.

3. The experiments and analyses conducted on each component of the proposed framework are comprehensive. Beyond demonstrating the feasibility of pretraining on desktop data, the paper offers a practical reference and a solid foundation for developing future learning systems capable of utilizing any video-based data for embodied intelligence.

4. The significant performance gains observed on downstream tasks following D2E pretraining clearly demonstrate the framework’s effectiveness in extracting generalizable knowledge from desktop data, highlighting its potential as a scalable and efficient pretraining paradigm.

**Weaknesses:**

1. The utilization of data with machine-generated pseudo labels leads to inconsistent performance changes across the Manipulation and Navigation benchmarks. This inconsistency suggests that the quality and reliability of pseudo labels may vary significantly depending on the task type, and further analysis is needed to understand their impact on downstream performance.

2. Conducting experiments solely on the Libero benchmark is insufficient to support the claim that pretraining on desktop data contributes to learning generalized embodied knowledge. Validation on real-world manipulation tasks would substantially strengthen the paper’s argument.

## Minor Issue

1. The font style used in the paper is inconsistent with the official ICLR template, and should be adjusted to comply with the formatting standards.

**Questions:**

1. Could the authors provide more details about the baseline implementations reported in Table 1? It would be helpful to clarify whether these baselines were reimplemented under the same settings or adopted from existing works.

2. There remains a considerable gap between game data and embodied manipulation data due to the coarse-grained differences in action spaces and the varying requirements for 3D physical understanding. A more comprehensive analysis is needed to quantify how much embodied manipulation tasks actually benefit from desktop data pretraining, and under what conditions such transfer is most effective.

**Details Of Ethics Concerns:**

There are several potential concerns regarding responsible research practices during human data collection and the legal compliance of using publicly sourced data. While the authors have provided an ethics statement in the Appendix, which addresses some of these issues, additional clarification may still be needed. In particular, the data privacy of the collected game videos should not be overlooked, as many gameplay recordings may inadvertently contain personal information, such as usernames, chat histories, or identifiable content. A clearer explanation of how such sensitive information is detected, filtered, or anonymized would strengthen the paper’s ethical rigor.

---

> ### Author Response · Authors · 2025-11-25
>
> > The utilization of data with machine-generated pseudo labels leads to inconsistent performance changes across the Manipulation and Navigation benchmarks. This inconsistency suggests that the quality and reliability of pseudo labels may vary significantly depending on the task type, and further analysis is needed to understand their impact on downstream performance.
>
> We thank the reviewer for this important observation. Our current analysis of pseudo-label effectiveness is based on empirical results (Tables 6–7), which show a decrease in manipulation performance (−4.4pp) and an improvement in navigation performance (+8.0pp).
>
> We hypothesize this task-dependent pattern arises from:
>
> * **Action space granularity**: Manipulation requires fine-grained, continuous control (e.g., precise gripper positioning), whereas navigation relies on coarser waypoint-level commands. As a result, pseudo-label noise is more detrimental to tasks that demand high precision.
> * **Domain gap**: Many desktop games (especially strategy, exploration, and FPS games) share more structural similarity with navigation tasks—such as spatial reasoning and waypoint planning—than with manipulation tasks, which involve 3D physical interaction and contact dynamics.
> * **Label noise tolerance**: Navigation benefits from diverse spatial reasoning patterns present across 20 games, where the scale and diversity of pseudo-labeled data can outweigh imperfections in label quality. Manipulation tasks, however, are more sensitive to action quality due to their tighter control requirements and lower tolerance for noise.
>
> While these explanations are consistent with our empirical trends, we acknowledge that a more controlled analysis would further clarify the underlying mechanisms. We are currently conducting additional real-world manipulation and navigation experiments to better understand how pseudo-label quality influences downstream performance, and we will share the corresponding preliminary results during the rebuttal period.

---

> ### Author Response · Authors · 2025-11-25
>
> > Conducting experiments solely on the Libero benchmark is insufficient to support the claim that pretraining on desktop data contributes to learning generalized embodied knowledge. Validation on real-world manipulation tasks would substantially strengthen the paper’s argument.
>
> We thank the reviewer for pointing out that evaluating generalization solely through LIBERO is insufficient for supporting the broader claim of desktop-to-embodied transfer. We agree that while LIBERO is a useful proxy, stronger evidence from more established or real-world manipulation settings is needed to substantiate generalized embodied knowledge.
>
> **1. Evaluation on Meta-World [6]**
>
> To address the concern regarding manipulation tasks, we evaluated our VAPT models on **Meta-World**, a standard benchmark for multi-task robotic manipulation. We compared VAPT against the InternVL3-1B baseline across tasks of varying difficulty.
>
> **Table: Success Rates on Meta-World Benchmark**
>
> | **Method** | **Easy** | **Medium** | **Hard** | **Very Hard** | **Avg** |
> | --- | --- | --- | --- | --- | --- |
> | Baseline (InternVL3-1B) | 55.4 | 14.5 | 1.7 | 8.0 | **19.9** |
> | VAPT (w/o pseudo) | 53.6 | 18.2 | 8.3 | 20.0 | **25.0** |
> | VAPT (w/ pseudo) | 52.1 | 16.4 | 6.7 | 24.0 | **24.8** |
>
> Implementation Details:
>
> To ensure a fair comparison, we used the official LeRobot v0.4.1 codebase. We adapted the InternVL3-1B backbone following the exact protocol used for SmolVLA, without modifying any architecture parameters. Both the baseline and VAPT models were trained for 50k steps with a learning rate of 1e-4, keeping all other hyperparameters at default settings. We conducted 10 evaluation episodes per task.
>
> Analysis:
>
> Even without robotics-specific pretraining or extensive hyperparameter tuning, VAPT consistently outperforms the baseline, showing an average success rate improvement of roughly 5% (a ~25% relative gain). This gap is most pronounced in the Hard and Very Hard categories (e.g., 8.0% vs. 20.0–24.0% on Very Hard), suggesting that the priors learned from desktop data are particularly robust for complex manipulation challenges.
>
> **2. Real-World Validation on SO-101 Arm**
>
> Beyond simulation, we are actively validating VAPT on a physical SO-101 robotic arm.
>
> - **Qualitative Results:** We have uploaded a demo video to the **anonymized website** linked in the abstract. This video demonstrates the model's capability to transfer desktop-learned priors to physical hardware.
> - **Quantitative Evaluation:** We are currently finalizing the quantitative metrics for these real-world tasks and will update the paper and the anonymous website with these results within the next few days.
>
> **3. Visualizing the Transfer Mechanism**
>
> Finally, to address the question of *why* desktop data is effective for manipulation despite the domain gap, we are currently working on visualizing the latent representations of our model. We aim to empirically show the alignment of sensorimotor primitives between desktop and robot trajectories in the feature space. We plan to update these visualization results to the paper within the next few days to provide further intuition on the transfer mechanism.
>
> We believe the combination of Meta-World results, real-world demonstrations, and the upcoming latent analyses addresses the concerns regarding benchmark reliability and the validity of desktop-to-robot transfer.
>
> Reference:
>
> [6] Yu et al., "Meta-World: A Benchmark and Evaluation for Multi-Task and Meta Reinforcement Learning," CoRL, 2019.

---

> ### Author Response · Authors · 2025-11-25
>
> > Could the authors provide more details about the baseline implementations reported in Table 1? It would be helpful to clarify whether these baselines were reimplemented under the same settings or adopted from existing works.
>
> Thank you for requesting this clarification.
>
> For the baselines reported in Table 1:
>
> * All baselines were reimplemented under the same experimental settings using our unified evaluation framework.
> * We matched input modalities, batch sizes, throughput measurement protocols, and hardware settings across all baselines to ensure a fair and consistent comparison.
> * No external throughput numbers were copied from prior work; all results were reproduced internally for consistency and reproducibility.
>
> **Correction to Benchmark Results:**
>
> We identified that unintended excessive disk I/O was contaminating throughput measurements. All models were re-evaluated using a temporary filesystem to eliminate this issue. Table 1 values have been updated accordingly.
>
> We have updated Section 3 and Appendix A.7–A.8 to include these implementation details and the corrected baseline results explicitly, and we will highlight these revisions for easy reference.
>
> ---
>
> > There remains a considerable gap between game data and embodied manipulation data due to the coarse-grained differences in action spaces and the varying requirements for 3D physical understanding. A more comprehensive analysis is needed to quantify how much embodied manipulation tasks actually benefit from desktop data pretraining, and under what conditions such transfer is most effective.
>
> We appreciate the reviewer’s insightful question and agree that a more comprehensive analysis is needed. We recognize that the reviewer's concern can be addressed through two complementary approaches: (1) controlled ablations to isolate specific factors (e.g., action space dimensionality, temporal alignment), and (2) validation across diverse manipulation tasks to establish the breadth of transfer.
>
> While both directions are valuable, our work has focused primarily on the latter. We believe that demonstrating the empirical feasibility of desktop-to-embodied transfer across multiple benchmarks provides the necessary foundation upon which deeper causal analysis can be built.
>
> **Validation Across Multiple Benchmarks:**
>
> As detailed in our response to Weakness 2, we have evaluated VAPT on **Meta-World**, a standard multi-task manipulation benchmark, in addition to LIBERO and CANVAS. VAPT improves the average success rate by roughly 5 percentage points (approximately 25% relative gain) over the InternVL3-1B baseline, with the largest improvements on Hard and Very Hard tasks. We are also conducting real-world validation on a physical SO-101 robotic arm, with a demo video available on the anonymized website and quantitative results to be shared during rebuttal. Please refer to our response to Weakness 2 for full implementation details and analysis.
>
> We acknowledge that controlled ablations to rigorously quantify the contribution of individual factors—such as action space dimensionality, temporal alignment, and domain similarity—would provide deeper insight into the conditions under which transfer is most effective. We view such analysis as important future work.
>
> Reference:
>
> [6] Yu et al., "Meta-World: A Benchmark and Evaluation for Multi-Task and Meta Reinforcement Learning," CoRL, 2019.

---

> ### Author Response · Authors · 2025-11-25
>
> > The font style used in the paper is inconsistent with the official ICLR template, and should be adjusted to comply with the formatting standards.
>
> Thank you for pointing this out. We have identified and removed the conflicting packages that caused font inconsistencies with the official ICLR template. The revised manuscript now fully complies with the formatting standards. We appreciate your attention to detail.
>
> > There are several potential concerns regarding responsible research practices during human data collection and the legal compliance of using publicly sourced data. While the authors have provided an ethics statement in the Appendix, which addresses some of these issues, additional clarification may still be needed. In particular, the data privacy of the collected game videos should not be overlooked, as many gameplay recordings may inadvertently contain personal information, such as usernames, chat histories, or identifiable content. A clearer explanation of how such sensitive information is detected, filtered, or anonymized would strengthen the paper’s ethical rigor.
>
> We are grateful to the reviewer for bringing forward this important concern. We take data privacy seriously and have implemented several safeguards to ensure responsible data collection, processing, and usage.
>
> **Human-Collected Data:**
> * All human demonstrations were collected from consenting participants who were explicitly informed about data usage
> * Participants were instructed to avoid displaying personally identifiable information during recording sessions
> * We conducted a manual review of collected data to identify and remove any inadvertent personal information
>
> **YouTube Pseudo-Labeled Data:**
> * We exclusively selected gameplay videos with permissive licenses (Creative Commons or equivalent) that explicitly allow derivative use
> * We focused on gameplay footage that naturally excludes personal information—specifically, we avoided streams with webcam overlays, voice chat, or personal commentary
> * The pseudo-labeling process operates on visual frames and predicted actions only, without capturing or storing any metadata, usernames, or chat content
>
> **Additional Safeguards:**
> * Our OWA Toolkit is designed to capture only screen content and input events, explicitly excluding system-level information, file paths, or application data outside the game window
> * We will expand our ethics statement in the camera-ready version to include these specific privacy protection measures
>
> We believe these measures adequately address privacy concerns while enabling reproducible research. We are committed to responsible data practices and welcome any additional suggestions from the reviewer.

---

### Author Response · Authors · 2025-12-02
**Result of Real-World Validation using SO-101 Arm**

To address the reviewers' concerns about relying solely on simulation-based manipulation tasks (Reviewers 5KYr, cyb3, SNK1), we additionally ran a real-world pick-and-place experiment. We used an SO101 robot arm and collected demonstrations from a hired teleoperator, who controlled a leader arm while the follower arm executed the manipulation. The task is specified by a single language instruction, "Pick the blue cube and place it in the white box," with the white box fixed and the blue cube placed at five different initial positions. This setup closely follows the real-world evaluation protocol used in SmolVLA[1].

We used the lerobot framework for data collection, training, and evaluation. To ensure a well-controlled and reproducible setup, we (i) used two cameras (top and side views), (ii) placed the scene against a green background/screen, (iii) fixed a carefully chosen initial rest pose during data collection, and (iv) randomized the evaluation order over models. In total, we collected 208 episodes and trained each model (baseline and VAPT variants) on this dataset, following the same downstream-task training protocol as in our other experiments.

For evaluation, each model was tested 10 times (2 trials for each of the 5 positions), for a total of 30 rollouts. We measure success as correctly picking the blue cube and placing it inside the white box. The resulting success rates are:

| Method                   | Success Rate (%) |
|--------------------------|------------------|
| Baseline (InternVL3-1B)  | 70.0             |
| VAPT (w/o pseudo-labels) | 80.0             |
| VAPT (w/ pseudo-labels)  | 80.0             |

We have already included a demo video on the anonymous project website, which clearly illustrates the experimental setup, and we will update the paper to include this real-world experiment. **These real-world results show that VAPT consistently improves over the baseline, mirroring the gains observed in simulation and supporting our claim that VAPT's benefits are not limited to sim-only settings.**

Reference:

[1] Shukor et al., "Smolvla: A vision-language-action model for affordable and efficient robotics," Arxiv 2025

---

### Author Response · Authors · 2025-12-03
**General Response**

Dear Reviewers and Area Chair,

We sincerely thank you for your constructive feedback and time dedicated to reviewing our work.

As the reviewers noted, exploring alternative data sources for embodied AI, given the high cost of physical robot data collection, is a meaningful research direction. Our proposed D2E (Desktop to Embodied AI) framework demonstrates that desktop interactions, particularly gaming data, can serve as a scalable pretraining substrate for robotics. We appreciate the reviewers' recognition of our contributions across multiple dimensions, including: the comprehensive end-to-end pipeline (Reviewers 5KYr, cyb3), the high-quality datasets and data collection infrastructure (Reviewer 5KYr), the solid engineering contributions (Reviewers cyb3, rEUi), and the significant performance gains on downstream tasks (Reviewer 5KYr).

In this rebuttal, we have carefully addressed all reviewer comments and substantially improved the manuscript with new experiments, analyses, and clarifications. The key updates include:

**New Experimental Results:**

- **Ablation on temporal offset τ:** We conduct a systematic study on the temporal offset parameter τ with τ ∈ {0, 50, 100, 150, 200} ms across six games, demonstrating that performance becomes relatively consistent once τ ≥ 100 ms.
- **Training loss curve analysis (Appendix E.1):** We compare training dynamics between VAPT and the baseline on LIBERO and CANVAS. Although we initially planned latent representation visualizations, we found that training dynamics provide clearer evidence for the transfer mechanism. VAPT shows immediate convergence without the initial plateau observed in the baseline, indicating that desktop-pretrained representations provide a better starting point for embodied control.
- **Evaluation on Meta-World benchmark:** VAPT achieves approximately 25% relative improvement over baseline, with particularly strong gains on Hard and Very Hard tasks, consistent with the improvements observed on LIBERO and CANVAS.
- **Revised data-loading baselines:** All data loading baselines in Table 1 were re-evaluated after identifying and fixing disk I/O contamination issues.
- **Real-world validation on physical SO-101 robotic arm:** In a pick-and-place task with 208 collected episodes and 30 evaluation rollouts, VAPT achieves an 80% success rate compared to a 70% baseline, confirming that desktop pretraining benefits transfer to physical robots.

**Enhanced Discussions:**

- **Expanded motivation for desktop-to-embodied transfer:** We detail concrete cost comparisons showing the scalability advantage of desktop data. Our approach required 14 annotators under 1 manager for 1 month (335 hours) vs. DROID[1]'s 50 collectors across 13 institutions over 12 months (350 hours).
- **Three key factors enabling desktop-to-embodied transfer:** (1) action modality alignment, (2) goal-directed sequential decision-making skills, (3) high diversity across visual, behavioral, and rule dimensions.
- **Task-dependent pseudo-label effectiveness:** Manipulation tasks show sensitivity to label quality (−4.4pp with pseudo-labels) while navigation tasks benefit from increased data diversity (+8.0pp), attributed to differences in action space granularity and domain similarity.
- **Design philosophy of NEP-τ:** We intentionally chose simplicity to fully exploit pretrained MLLM without architectural modifications; NEP-τ is a natural and minimal extension of standard NEP (τ=0) for inverse dynamics modeling.
- **Clarification on pretraining data:** VAPT pretraining uses only desktop game data without any robotics datasets; robotics data (LIBERO, CANVAS, Meta-World, real-world) is used exclusively for task-specific finetuning.
- **Expanded ethics statement (Appendix I):** We provide additional details on privacy and data handling for both human-collected and YouTube-sourced data.

In the revised manuscript, all revisions are temporarily highlighted in **blue** for ease of review.

We hope our responses and revisions fully address the reviewers' concerns. We believe D2E makes substantial contributions to the community through: (1) establishing the first complete pipeline from scalable desktop data collection to verified transfer in embodied domains, (2) demonstrating strong empirical results (96.6% on LIBERO manipulation, 83.3% on CANVAS navigation), and (3) our commitment to open science—we will release the OWA Toolkit, all datasets (335 hours human-collected + 1K+ hours pseudo-labeled), and VAPT-trained models to enable reproducible research.

Thank you again for your valuable feedback and constructive insights.

Best regards,\
Authors

Reference:

[1] Khazatsky et al., "DROID: A Large-Scale In-The-Wild Robot Manipulation Dataset," RSS, 2024.

---

### Meta-Review · Area_Chair_iqKs · 2026-01-07

**Summary:**

This work demonstrates that large-scale desktop interaction data as a practical pretraining source for embodied AI, improving real-world manipulation and navigation performance. The paper receives initial scores of 8/6/4/4; during rebuttal, one reviewer with an initial score of 4 indicates an intention to raise the score, while the remaining 4 does not respond. Given the positive overall evaluations, I am inclined to accept this paper as a poster.

**Reviewer Concerns:**

1. **Motivation and significance of desktop-to-embodied transfer:** The authors address the concern of the motivation for why desktop data is a meaningful and scalable alternative to physical robot data. They also mentioned three key factors enabling desktop-to-embodied transfer: (1) action modality alignment, (2) goal-directed sequential decision-making skills, (3) high diversity across visual, behavioral, and rule dimensions.

2. **Data quality, scale, and baselines:** The authors clarify the data sources, fix and re-evaluate data-loading baselines, and explain the role and limitations of pseudo-labeled data across different task types.

**Reviewer Scores:**

Initial Score:
5KYr: 8, cyb3: 6, SNK1: 4, rEUi: 4

After Rebuttal:
5KYr: 8, cyb3: 6, SNK1: 4 (would like to improve), rEUi: 4

---

### Decision · Program_Chairs · 2026-01-26

Accept (Poster)